# Integrating Tree Path in Transformer for Code Representation

**Han Peng**[1], **Ge Li**[1*], **Wenhan Wang**[1], **Yunfei Zhao**[1], **Zhi Jin**[1*]

[1]Key Laboratory of High Confidence Software Technologies (Peking University),
Ministry of Education; Institute of Software, EECS, Peking University, Beijing, China
{phan, lige, wwhjacob, zhaoyunfei, zhijin}@pku.edu.cn

## Abstract

Learning distributed representation of source code requires modelling its syntax and semantics. Recent state-of-the-art models leverage highly structured source code representations, such as the syntax trees and paths therein. In this paper, we investigate two representative path encoding methods shown in previous research work and integrate them into the attention module of Transformer. We draw inspiration from the ideas of positional encoding and modify them to incorporate these path encoding. Specifically, we encode both the pairwise path between tokens of source code and the path from the leaf node to the tree root for each token in the syntax tree. We explore the interaction between these two kinds of paths by integrating them into the unified Transformer framework. The detailed empirical study for path encoding methods also leads to our novel state-of-the-art representation model TPTrans, which finally outperforms strong baselines. Extensive experiments and ablation studies on code summarization across four different languages demonstrate the effectiveness of our approaches. We release our code at `https://github.com/AwdHanPeng/TPTrans`.

## 1 Introduction

Machine learning for source code aims at building models that learn semantic embedding of programs. The initial representation of source code relied on sequential models adopted from natural language processing, such as $n$-gram language model [Hindle et al., 2016, Hellendoorn and Devanbu, 2017], Recurrent Neural Networks (RNNs) [Wei et al., 2019] and so on. However, source code is more logical than natural languages, rich in structured information such as Abstract Syntax Tree (AST). Therefore, these previous works struggle to capture the structural complexity of source code.

Some works leverage the program AST to model source code structure and linearize the graph by traversing it. Code2seq [Alon et al., 2018] represents source code as a set of pairwise paths over AST where each path is compressed to a vector using LSTMs [Hochreiter and Schmidhuber, 1997]. Code2seq obtains state-of-the-art for code representation using only AST information, demonstrating the effectiveness of path encoding. [Kim et al., 2020] leveraged another kind of path from the leaf node to the root of AST by traversing up its ancestors and then coupled the representation of paths with the source code.

On the other hand, several works also leverage structured graph neural networks for modelling source code directly. [Allamanis et al., 2017] used Gated Graph Neural Network (GGNN) to embed the semantically meaningful relationships in the code. [Zhou et al., 2019] proposed a novel graph neural network with composite programming representation for vulnerability identification. However, GNNs

---

*Corresponding author

35th Conference on Neural Information Processing Systems (NeurIPS 2021).

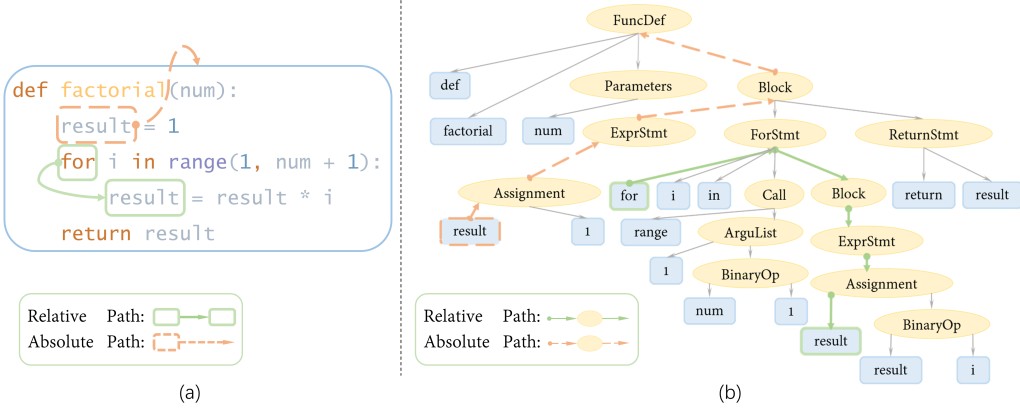

Figure 1: Example of (a) a python code snippet and (b) the syntax tree of it. The relative path across the syntax tree between tokens **for** and **result** represents the relationship between them, showing the pattern of for loop in the code snippet. Meanwhile, the absolute path for token **result** starts at the leaf node and ends at the tree root, which reveals the behaviour of assignment for this token.

typically rely on synchronous message passing, requiring message passing iterations to aggregate information [Allamanis et al., 2017, Fernandes et al., 2018]. Besides, primarily for computational reasons, GNNs for programs usually compute few message-passing iterations. For these reasons, the representation of GNNs tends to be local and struggle to leverage long-range interactions.

On the contrary, Transformer [Vaswani et al., 2017] allows global information flow at each step but lacks the ability to model the structural complexity of source code. Recently, some works have explored introducing structural inductive bias in Transformer to access the global structural representation of source codes. [Hellendoorn et al., 2019] proposed Graph Relation Embedding Attention Transformer (GREAT), which extends [Shaw et al., 2018] and biases Transformer with relational information from graph edge types. After that, [Zügner et al., 2021] proposed Code Transformer based on XLNet [Yang et al., 2019], which computes pairwise distances on AST and integrate multiple relations into the attention module.

In this paper, we pursue the research line of combining Transformer with additional structure information of source code. Our first starting point is the structural model of Code2seq [Alon et al., 2018], which obtains state-of-the-art of code summarization using only pairwise path information in AST. However, Code2seq lacks the modelling of context, which leads us to explore combining path representation with source code context. The other starting point is Code Transformer [Zügner et al., 2021], which counts node distances cross AST to capture source code structure. However, the different nodes combinations of paths contain plenty of structure information, which is overlooked by only encoding distances.

To overcome the drawback shown in previous works, we adopt the ideas of encoding path to represent source code and integrate them to bias the attention module of Transformer. Specifically, we encode the path between source code tokens across AST and the path from leaf node to tree root for each token. Then we draw inspiration from both *relative* and *absolute positional* encoding methods in the NLP field and modify them to integrate *path* encodings into Transformer. These paths introduce inductive bias into the attention module of Transformer, powering it to know the structure of source codes. For clarity, we name the first path as the *relative path* and the last one as the *absolute path* (see Fig1), since much similarities between *path* and *position* on *relative* or *absolute* encoding exist.

Intuitively, the relative path represents the structural relationship between tokens and shows specific patterns of code block such as `For` loop, etc. Meanwhile, the absolute path complements the structural information for each token and reveals the program behaviour on it, such as `Assignment` and `Call`, etc. Both two kinds of paths contain plenty of structural information of code syntax. Actually, lots of works have introduced these paths into many novel models [Kim et al., 2020, Alon et al., 2018, 2020]. However, to our best knowledge, the feature relationship between these two paths for code representation learning is still not completely studied. In this paper, we integrate these paths into the unified Transformer framework to analyze each effectiveness and their relationship. The detailed

empirical study also leads to our novel code representation model TPTrans, which means encoding the **T**ree **P**ath into the **Trans**former. We show the effectiveness of our approaches on the code summarization task across four different languages datasets, in which the model predicts a method's name based on its body. Our model finally outperforms solid baselines and obtains state-of-the-art on most datasets. The contributions of this paper are summarized as follows:

1. We propose TPTRANS which integrates path encoding in Transformer, powers it to know the structure of source code, and our model significantly outperforms existing baselines.

2. We present the empirical study to investigate the relationship between two kinds of path encoding methods proposed in previous works, shedding light on this line of future work.

## 2 Related work

**Representation learning for source code**   The growing availability of open-source code creates opportunities for machine learning from big code. Early research works learned language model on raw text data for code [Dam et al., 2016, Wang et al., 2016, Allamanis et al., 2016, Iyer et al., 2016], providing evidence for the naturalness assumption [Hindle et al., 2016]. These models treat codes as sequences rather than trees and struggle to model the structural complexity of source code. After that, some works leveraged structural information to model source code. [Mou et al., 2016] proposed tree-based convolutional neural networks to represent source code. [Alon et al., 2018, 2019] treated source code as weighted combination of pairwise paths in abstract syntax trees. Several works also use structured graphical models to represent structural aspects of source code directly. For example, [Allamanis et al., 2017] proposed GGNN to represent program graphs consisting of AST together with control-flow and data-flow edges. After that, [Hellendoorn et al., 2019, Zügner et al., 2021] proposed to bias the self-attention computation of Transformer given the underlying graph structure. In this paper, we explore integrating path representation of syntax tree into Transformer, powering it to capture the structure of code snippets.

**Structural representation of languages**   It has been proved that modelling the language structure helps to represent the compositional aspects of natural language [Socher et al., 2011] and improves the generalization of models [Chen et al., 2016]. Several works [Shen et al., 2018, Shiv and Quirk, 2019] proposed new architectures for modeling tree structures, including in the domain of natural languages processing [Wang et al., 2019] and code representation learning [Hellendoorn et al., 2019, Zügner et al., 2021]. We pursue this research line and show the benefits of integrating tree structure as inductive bias.

**Positional encoding for Transformer**   The absolute positional encoding used in Transformer has been proved inefficient to model the order information of sequence. [Shaw et al., 2018] firstly proposed to leverage relative positional encoding to Transformer. After that, Transformer-XL [Dai et al., 2019] re-parameterized the self attention of Transformer to integrate relative positional encoding and T5 [Raffel et al., 2019] simplified the vector representation of relative positions in [Shaw et al., 2018] to scalars. Recently, several works also tried to enhance positional encodings in Transformer to integrate structural inductive bias. For example, [Shiv and Quirk, 2019] extended the sequential positional encoding to tree-based positional encoding in Transformer, and [Zügner et al., 2021] modified the relative positional encoding of Transformer-XL to incorporate multiple relations in tree structural. In this paper, we adopt the idea to modify positional encodings of Transformer and introduce structural bias of path information in the syntax tree for self-attention computing.

## 3 Approaches

Self-attention is one of the key components of Transformer. The attention module can be formulated as querying a dictionary with key-value pairs. The matrix form equation for attention with a single head is

$$Attention(Q, K, V) = softmax(\frac{QK^T}{\sqrt{d}})V, \tag{1}$$

where $d$ is the dimension of the hidden representations, and $Q$(Query), $K$(Key), $V$(Value) are specified as the hidden representations of the previous layer. For simplicity, we omit the index of

layer, denote $x = (x_1, x_2 \cdots, x_n)$ and $z = (z_1, z_2 \cdots, z_n)$ as the input and output of self-attention module in the same layer respectively, where $n$ is the sequence length. Then Eq.1 can be rewritten as

$$z_i = \sum_{j=1}^{n} \frac{\exp{(\alpha_{ij})}}{\sum_{j'=1}^{n} \exp{(\alpha_{ij'})}} (x_j W^V), \ where \ \alpha_{ij} = \frac{1}{\sqrt{d}} (x_i W^Q)(x_j W^K)^T, \quad (2)$$

where $W^Q \in \mathbb{R}^{d \times d_k}$, $W^K \in \mathbb{R}^{d \times d_k}$, $W^V \in \mathbb{R}^{d \times d_v}$ and we set $d_k = d_v = d$. Eq.2 is oblivious to structured input since it views input as unordered vector sets. In the NLP field, several works [Shaw et al., 2018, Ke et al., 2020, Raffel et al., 2019, Dai et al., 2019] propose different ways to bias Transformer towards sequential inputs by incorporating positional information into the self-attention module. Generally, there are two categories of positional encoding methods for self-attention: absolute positional encoding and relative positional encoding.

The original Transformer [Vaswani et al., 2017] use absolute positional encoding to represent positions. Given a sentence, a real-valued vector $p_i \in \mathbb{R}^d$ will be added to the word embedding $w_i$ at position $i$, and the sum vector will be used as the input to Transformer. [Ke et al., 2020] further proposed TUPE to untie the correlations between absolute positions and words and directly model the relationship between a pair of words or positions using different projection matrices, that is

$$\alpha_{ij} = \frac{1}{\sqrt{2d}} (x_i W^Q)(x_j W^K)^T + \frac{1}{\sqrt{2d}} (p_i U^Q)(p_j U^K)^T, \quad (3)$$

where $U^Q, U^K \in \mathbb{R}^{d \times d}$ are projection matrices for positional embedding and scaling term $\sqrt{2d}$ is used to retain the magnitude of $\alpha_{ij}$.

Using different $p_i$ helps Transformer distinguish the words at different positions $i$. However, the absolute positional encoding is not effective in capturing the relative word orders. Therefore, [Shaw et al., 2018] proposed the relative positional encoding, that is

$$z_i = \sum_{j=1}^{n} \frac{\exp{(\alpha_{ij})}}{\sum_{j'=1}^{n} \exp{(\alpha_{ij'})}} (x_j W^V + a_{ij}^V), \ where \ \alpha_{ij} = \frac{1}{\sqrt{d}} (x_i W^Q)(x_j W^K + a_{ij}^K)^T \quad (4)$$

where $a_{ij}^V, a_{ij}^K \in \mathbb{R}^d$ are learnable parameters and can be viewed as the relative position embedding between position $i$ and $j$. Eq.4 extends self-attention to consider the pairwise positional relationship and models it as a directed edge between head and tail word entities [Shaw et al., 2018], which means the tail entity should be close to the embedding of the head entity plus vector depending on the relationship [Bordes et al., 2013].

We draw inspiration from the ideas of encoding absolute and relative positional information and modify Eq.3 and Eq.4 to incorporate path information in the syntax tree of source code.

### 3.1 Integrating path encodings in attention module

A syntax tree uniquely represents a source code snippet in the given language and grammar. In a syntax tree, the leaves are called terminals, and the non-leaf nodes are called nonterminals. [Alon et al., 2018, 2019] used AST to represent a source code snippet. They considered pairwise paths between terminals, representing them as sequences of the terminal and nonterminal nodes in AST. [Kim et al., 2020] used another kind of path from the terminal to AST root by traversing up its ancestors, and then coupled paths representation with tokens and fed the sum representation into Transformer.

We adopt the above methods of leveraging paths and integrate them into the self-attention module of the unified Transformer framework. We name the path between terminals as the relative path and the path from terminal to root as the absolute path. Technically, we modify the *positional* encoding methods in the NLP field to integrate *path* encoding into Transformer, which shows natural analogies from the *positional* to *path* encoding ways as follows.

All kinds of paths are composed of nonterminals from a limited vocabulary of syntax grammar. We use a learned embedding matrix to convert nonterminals to vectors first.

**Relative path encoding**   The embed vectors of a relative path are first fed into a sequence encoder. We use a bi-directional GRU [Cho et al., 2014] to encode the path and use the final state to represent it, that is

$$r_{ij} = GRU_r(Path_{x_i \to x_j}), \tag{5}$$

where $Path_{x_i \to x_j}$ is the nonterminal vector sequence of the relative path between tokens $x_i$ and $x_j$, and $r_{ij} \in \mathbb{R}^d$ is the final vector representation of this path. Specifically, $Path_{x_i \to x_j}$ is a vector list $[n_0, n_1, \ldots, n_m]$, where $m$ is the path length and each $n$ is the embedding vector of each nonterminal node which is looked up from the embedding matrix of node type.

The pairwise path across the syntax tree reveals the structural relationship between two code tokens. Similarly, Eq.4 models the positional relationship between two words, so it is natural to modify the Eq.4 to integrate path encoding $r_{ij}$ into self-attention module. We firstly integrate $r_{ij}$ into the Query-Key product, that is

$$\alpha_{ij} = \frac{1}{\sqrt{d}}[(x_i W^Q)(x_j W^K + \underbrace{r_{ij} W_r^K}_{1})^T], \tag{6}$$

and then we integrate $r_{ij}$ into the weighted sum of Value, that is

$$z_i = \sum_{j=1}^{n} \frac{\exp(\alpha_{ij})}{\sum_{j'=1}^{n} \exp(\alpha_{ij'})}(x_j W^V + \underbrace{r_{ij} W_r^V}_{2}), \tag{7}$$

where $W_r^K, W_r^V \in \mathbb{R}^{d \times d}$ are projection matrices of Key and Value for $r_{ij}$, respectively. For clarity, we label the modifications for Key and Value as 1 and 2, respectively.

**Absolute path encoding**   The embed vectors of an absolute path are fed into another bi-directional GRU [Cho et al., 2014], that is

$$a_i = GRU_a(Path_{x_i \to root}), \tag{8}$$

where $Path_{x_i \to root}$ is the nonterminals vectors sequence of the absolute path from tokens $x_i$ to the syntax tree root, and $a_i \in \mathbb{R}^d$ is the final vector representation of this path.

The absolute path for each source code token can be viewed as a directed edge connecting the terminal with the tree root, which means the absolute path can calibrate the coordinates for each terminal in the special *structural space* that assigns the tree root as the origin. So it is natural to modify the absolute positional encoding in Eq.3 to integrate path vectors into the self-attention module, that is

$$\alpha_{ij} = \frac{1}{\sqrt{d}}[(x_i W^Q)(x_j W^K)^T + (a_i W_a^Q)(a_j W_a^K)^T], \tag{9}$$

where $W_a^Q, W_a^K \in \mathbb{R}^{d \times d}$ are projection matrices of Query and Key for absolute path representation, respectively.[2]

We think it is meaningful to investigate the feature interaction between absolute and relative paths and focus on both of them instead of overlooking either. So we further combine the relative and absolute path encoding shown in previous paragraphs and merge Eq.6 and Eq.9 as

$$\alpha_{ij} = \frac{1}{\sqrt{d}}[(x_i W^Q)(x_j W^K + \underbrace{r_{ij} W_r^K}_{1})^T + (a_i W_a^Q)(a_j W_a^K)^T]. \tag{10}$$

In the paragraphs above, we propose several modifications to integrate paths into self-attention. To sum up, we combine them and propose two versions of models. We firstly name the method of only relative path encoding as TPTRANS (Eq.6 + Eq.7). Then, to analyze the relationship between the relative and absolute path encoding further, we name the method of combining both of them as TPTRANS-$\alpha$ (Eq.10 + Eq.7).

---

[2]We tried the method adding path representation with token directly like [Kim et al., 2020] but no improvement, which we think is similar to the conclusion of untying correlations in TUPE. Besides, we tried to keep the scaling term $\sqrt{2d}$ in Eq.3 but no improvement.

# 4 Experiment setup

Code summarization is one of the most popular tasks in code representation learning. Given a function body, this task is to predict the function name. As observed by [Allamanis et al., 2016, Alon et al., 2019], code summarization is a good benchmark for code representation learning, as the method body typically forms complete logical units and the method name tends to be descriptive and precise. In this task, our approaches jointly learn code contexts and path representation in AST and predict the target method name as a sequence of sub-tokens. We measure precision, recall, and F1 score over target sequences in the case insensitive like [Alon et al., 2018, Zügner et al., 2021].

**Datasets**   To show the effectiveness of our approaches across different source code languages, we experiment on four datasets introduced in the CodeSearchNet (CSN) Challenge [Husain et al., 2019]: Python, Javascript, Go, and Ruby. The datasets from CodeSearchNet have been carefully deduplicated to avoid data leakage from training sets. See Table 1 for a summary of the datasets.

Table 1: Dataset statistics

| Dataset | Samples per partition | | |
| | Train | Valid | Test |
| --- | --- | --- | --- |
| CSN-Python | 412,178 | 23,107 | 22,176 |
| CSN-Javascript | 123,889 | 8,253 | 6,483 |
| CSN-Ruby | 48,791 | 2,209 | 2,279 |
| CSN-Go | 317,832 | 14,242 | 14,291 |

**Preprocessing**   We produce the AST for each method using the open-source parser Tree-Sitter,[3] in which all the code tokens are natively mapped as terminals (see Fig1). We adopt the way to split sub-tokens following [Alon et al., 2018, Zügner et al., 2021], in which each code token is split into sub-tokens respective to code naming conventions, e.g., `setConnectionsPerServer` is split into [`set`, `connections`, `per`, `server`]. We limit the vocabulary of sub-tokens with at least 100 occurrences in training sets, remove all punctuations and restrict the max code length 512. We remove all anonymous Javascript functions following [Zügner et al., 2021]. We assign different ids for nonterminals and do not split the literal of them. We set the max path length 32 covering almost all paths and make padding for short paths. For paths longer than 32, we sample nodes with equal intervals to maintain max length instead of truncating the last nodes since we assume the nodes at both left and right ends of the path are equally important.

**Parameter sharing**   In our implementation, the embedding matrix for nonterminals is shared for both absolute and relative paths since both kinds of paths are made up of nonterminals within the same syntax tree. We do not share the path encoder GRU for two different kinds of paths. Since the number of nonterminal types is far less than the size of source code vocabulary, we set different embedding matrices for nonterminals and source tokens, respectively. For efficiency, we share in different layers and heads for the projection matrices $W_r^K, W_r^V$ in relative path encoding and $W_a^Q, W_a^K$ in absolute path encoding. We also share the path representation across different layers and heads, which means we pre-encode paths only once by GRU before feeding features into Transformer.

**Complexity optimization**   The naive way for relative path encoding needs to encode all the pairwise paths for each code snippet and to compose the relative path representation matrix $R \in \mathbb{R}^{n \times n \times d}$, where $n$ is code length and $d$ is the dimension of hidden representations. This approach costs $\mathcal{O}(n^2)$ complexity since the number of pairwise paths encoded is $\frac{n(n-1)}{2}$. To optimize it, we notice that much repetition exists in all relative paths of a sample.[4] That means we only need to encode all the *unique* pairwise paths for each sample. We record the mapping between token pairs and unique paths as matrix $M \in \mathbb{R}^{n \times n}$. After that, we compute the Query-Key product between tokens and unique paths, and `gather` the product scores using matrix $M$. After calculating attention distribution, we

---

[3]https://github.com/tree-sitter/

[4]For example, if we parse a function that has three formal arguments to a syntax tree, we could see that the pairwise relative path between any two arguments will be the same.

`scatter` attention probability to each unique path and weighted sum using matrix $M$ again. This new approach avoids encoding $\mathcal{O}(n^2)$ paths and reduces it into $\mathcal{O}(l)$ where $l$ is the number of unique paths and is far less than $n^2$. It also reduces matrix $R$ from $\mathcal{O}(n^2 d)$ to $\mathcal{O}(ld)$. In our experiment, we set the max $l$ 512 for the relative path, which covers most of the samples. Besides, the complexity of absolute path encoding is original $\mathcal{O}(n)$, but we also reduce it in the same way and set the max $l$ 256 for it. We refer readers to the appendix for further details.

**Normalization**  We integrate vector sequence of path nodes as the inductive bias for the attention module of Transformer, which is not just trivial technically. Firstly, directly integrating standard RNN into the deep backbone of Transformer most likely results in gradient exploding or vanishing. Secondly, the attention procedure of Transformer is similarity calculation between vectors. The $Q$, $K$, and $V$ of the attention module are normalized while standard RNN does not limit vector norm, leading to size instability for dot-product attention between word and path representation. To overcome these issues, we first replace the standard GRU as a layer-normalized GRU [Ba et al., 2016], getting more stable parameters updates. After getting the final state of GRU, we feed it into another normalization layer to keep it the same vector norm with $Q$, $K$, and $V$. The normalization is much efficient shown in the following ablation result.

**Hyperparameters**  We denote the number of layers for encoder and decoder as $L_E$ and $L_D$, the hidden size as $D$, the feed-forward dimension as $D_{FF}$, and the number of heads as $H$. We primarily report results on two model size for both TPTrans and TPTrans-$\alpha$: base ($L_D$=1, $D_{FF}$=2048) and large ($L_D$=3, $D_{FF}$=4096). For base and large settings, we all set $L_E$=3, $D$=1024, and $H$=8. We add a pointer network [Vinyals et al., 2015] to the decoder as same as our baselines. The base model setting keeps the same size as other baselines for fair comparisons. The embedding dimension of the word and path node are 512 and 64, and we apply a linear layer to project word embedding to the hidden size of Transformer. We use one layer Bi-GRU, set the hidden size to 64, and concat the final states of forward and backward as output with the size of the single head dimension. We set the batch size and dropout rate to 128 and 0.2 and employ label smoothing of 0.1. All models and baselines are trained from random initial parameters. As the optimizer, we use Adam [Kingma and Ba, 2014] with a learning rate and weight decay of $1e^{-4}$. In our experiment, we use 4 Tesla V100 GPUs for training.

**Baselines**  Combining two views of *context* and *structure* for source code has been widely discussed in Code Intelligence. The *context* mainly refers to representing code as a sequence of text, and the *structure* refers to extracting knowledge from code syntax. Here we mainly compare with Code2seq [Alon et al., 2018], Graph Relational Embedding Attention Transformer (GREAT) [Hellendoorn et al., 2019], XLNet [Yang et al., 2019] and Code Transformer [Zügner et al., 2021]. Unlike the Transformer-based baselines, Code2seq uses only pairwise path information in AST to represent code snippets, which means it mainly focuses on code structure within the syntax tree and lacks context modelling. On the contrary, our proposed models and other baselines are designed based on Transformer naturally focused on context. Specifically, GREAT uses the relative positional encoding present in [Shaw et al., 2018] to bias the attention via manually designed edges, and Code Transformer is a recent Transformer model based XLNet, incorporating multiple relations to learning both structure and context jointly. XLNet is designed initially for huge-corpus natural language pretraining, but its novel two-stream self-attention also shows powerful performance in code representation learning.

Table 2: Model comparison

| Model | Context | Structure |
|---|---|---|
| Code2Seq | No | Relative Path |
| Great | Yes | Manually Designed Edges |
| XLNet | Yes | No |
| CodeTransformer | Yes | Multiply Structural Distances |
| TPTrans | Yes | Relative Path |
| TPTrans-$\alpha$ | Yes | Relative and Absolute Path |

Essentially, the difference between our model and baselines GREAT and Code Transformer is encoding path vs encoding edges in GREAT and distances in Code Transformer. The different node combinations of paths naturally contain plenty of structure information, which is overlooked by encoding distances. See Table 2 for clear comparisons between our models and baselines.

# 5 Results

## 5.1 Overall comparison

Table 3: Code summarization result on the CSN dataset

| Model | Python | | | Ruby | | | Javascript | | | Go | | |
|---|---|---|---|---|---|---|---|---|---|---|---|---|
| | Prec. | Rec. | F1 | Prec. | Rec. | F1 | Prec. | Rec. | F1 | Prec. | Rec. | F1 |
| Code2seq | 35.79 | 24.85 | 29.34 | 23.23 | 10.31 | 14.28 | 30.18 | 19.88 | 23.97 | 52.30 | 43.43 | 47.45 |
| Great | 35.09 | 31.62 | 33.26 | 24.66 | 22.25 | 23.39 | 31.25 | 26.87 | 28.89 | 50.02 | 46.52 | 48.21 |
| XLNet | 37.39 | 32.01 | 34.49 | 29.88 | 25.89 | 27.74 | 33.33 | 26.86 | 29.75 | 51.79 | 47.58 | 49.60 |
| CodeTransformer | 36.41 | 33.68 | 34.99 | 31.46 | 24.50 | 27.55 | 35.07 | 29.65 | 32.13 | 55.09 | 48.05 | 51.33 |
| TPTrans | 38.45 | 33.63 | 35.88 | **32.70** | **27.75** | **30.02** | 33.47 | 28.27 | 30.65 | **56.19** | **51.14** | **53.54** |
| [Base] | ±0.17 | ±0.06 | ±0.05 | ±1.32 | ±0.38 | ±0.72 | ±0.62 | ±0.52 | ±0.04 | ±0.21 | ±0.56 | ±0.22 |
| TPTrans-$\alpha$ | **38.48** | **33.99** | **36.09** | 32.54 | 26.77 | 29.38 | **34.06** | **28.42** | **30.99** | 56.00 | 50.97 | 53.37 |
| [Base] | ±0.07 | ±0.29 | ±0.18 | ±0.15 | ±0.50 | ±0.35 | ±0.21 | ±0.12 | ±0.14 | ±0.33 | ±0.28 | ±0.22 |
| TPTrans | **38.76** | 34.66 | **36.59** | 32.40 | 27.63 | 29.82 | **33.83** | 28.37 | 30.86 | **55.79** | 51.05 | 53.30 |
| [Large] | ±0.49 | ±0.35 | ±0.02 | ±0.84 | ±0.78 | ±0.79 | ±0.55 | ±0.54 | ±0.53 | ±1.30 | ±1.30 | ±0.83 |
| TPTrans-$\alpha$ | 38.39 | **34.70** | 36.45 | **33.07** | **28.34** | **30.52** | 33.68 | **28.95** | **31.14** | 55.67 | **51.31** | **53.39** |
| [Large] | ±0.18 | ±0.26 | ±0.22 | ±0.86 | ±0.61 | ±0.53 | ±0.34 | ±0.54 | ±0.46 | ±0.92 | ±0.76 | ±0.13 |

The overall comparison results are shown in Table 3. We show the performance of two different settings for both TPTrans and TPTrans-$\alpha$. The base setting keeps the same Transformer backbone as other Transformer-based baselines, and the large setting additionally widens the feed-forward dimension and increases the number of decoder layers. We find that our proposed models TPTrans and TPTrans-$\alpha$ substantially outperform all baselines on all but one language, highlighting the effectiveness of learning path information. The only exception is Javascript, while our models still outperform most of the baselines on this dataset. Besides, we find that the large setting further improves the performance on most datasets.

It is worth mentioning that Code Transformer is designed based on XLNet, which natively equips the relative positional encoding of context and is effective for modelling long sequences. In this work, we mainly focus on studying the effectiveness of encoding path, so we use the original Transformer as our backbone. Even so, we still obtain advanced results on almost all datasets. We also find that compared to the improvement of Code Transformer over XLNet, our models get a more remarkable rise over Code Transformer.

To investigate the exception of Javascript dataset, we focus on TPTrans and count the mean and variance of all relative paths length across four datasets. We also count the total numbers of unique relative paths in training datasets and present them in Table 4.[5] We firstly find that the path length of Javascript is not only longer but also has a significantly larger variance than other datasets. We then find that the unique paths num of Javascript is much large and does not match its dataset size well, which likely indicates much noise exists in the training dataset. So we assume that due to such property of Javascript, it is harder to extract useful structural information from path than others. Accordingly, our models achieve more significant performance on Ruby dataset, and the Ruby dataset has smaller mean length and variance than others and also keeps an ideal num of unique paths. As for the outlier of Javascript, one possible reasoning line is the inherent property of Javascript language results in complicated syntax. We also suspect that it is due to the internal design of language parsers, and perhaps a more suitable parser might alleviate this problem.

Table 4: Path statistics

| Dataset | Train | | Valid | | Test | | Train | |
|---|---|---|---|---|---|---|---|---|
| | Mean | Var. | Mean | Var. | Mean | Var. | Path nums | Data size |
| Ruby | 9.29 | 17.62 | 9.76 | 18.84 | 9.60 | 18.82 | 10.97M | 48,791 |
| Go | 11.18 | 20.16 | 11.56 | 22.40 | 10.68 | 18.83 | 45.79M | 317,832 |
| Python | 11.96 | 24.61 | 11.81 | 23.64 | 12.05 | 25.36 | 81.42M | 412,178 |
| Javascript | **14.59** | **41.46** | **14.40** | **40.02** | **14.57** | **40.59** | 52.60M | 81,487 |

Lastly, we find that the overall performance of TPTrans and TPTrans-$\alpha$ does not differ so much, although the latter seems to obtain additional information about the absolute path. We investigate the effectiveness of both paths and analyze the relationship between them further in the following.

---

[5] For Javascript, we present the sample num of non-anonymous functions.

## 5.2 Empirical study

In this work, we propose two versions of models, namely TPTrans and TPTrans-$\alpha$. The first one encodes the pairwise relative paths in the syntax tree, while the last one additionally encodes the absolute path. Here we explore the feature interaction of relative and absolute path encodings and mainly experiment on the large model setting.

Table 5: Empirical study for path relationship

| Model | Python | | | Ruby | | | Javascript | | | Go | | |
|---|---|---|---|---|---|---|---|---|---|---|---|---|
| | Prec. | Rec. | F1 | Prec. | Rec. | F1 | Prec. | Rec. | F1 | Prec. | Rec. | F1 |
| TPTrans | **38.76** | **34.66** | **36.59** | **32.40** | **27.63** | **29.82** | **33.83** | **28.37** | **30.86** | **55.79** | 51.05 | **53.30** |
| | ±0.49 | ±0.35 | ±0.02 | ±0.84 | ±0.78 | ±0.79 | ±0.55 | ±0.54 | ±0.53 | ±1.30 | ±1.30 | ±0.83 |
| w/o *Rel* in $V$ | 37.00 | 33.33 | 35.06 | 29.96 | 25.87 | 27.76 | 31.93 | 27.31 | 29.43 | 54.48 | **51.56** | 52.97 |
| | ±0.38 | ±0.77 | ±0.46 | ±0.64 | ±0.90 | ±0.76 | ±0.71 | ±0.77 | ±0.66 | ±1.12 | ±0.68 | ±0.34 |
| w/o *Rel* in $K,V$ | 34.30 | 28.99 | 31.42 | 24.78 | 20.32 | 22.33 | 27.94 | 23.08 | 25.28 | 51.47 | 47.71 | 49.52 |
| | ±0.35 | ±0.44 | ±0.12 | ±0.51 | ±0.46 | ±0.48 | ±0.37 | ±0.41 | ±0.38 | ±0.34 | ±0.22 | ±0.10 |
| TPTrans-$\alpha$ | **38.39** | **34.70** | **36.45** | **33.07** | **28.34** | **30.52** | **33.68** | **28.95** | **31.14** | **55.67** | 51.31 | **53.39** |
| | ±0.18 | ±0.26 | ±0.22 | ±0.86 | ±0.61 | ±0.53 | ±0.34 | ±0.54 | ±0.46 | ±0.92 | ±0.76 | ±0.13 |
| w/o *Rel* in $V$ | 37.28 | 33.63 | 35.36 | 30.44 | 25.27 | 27.58 | 32.93 | 28.29 | 30.43 | 54.89 | **51.35** | 53.06 |
| | ±0.35 | ±0.44 | ±0.34 | ±1.47 | ±1.11 | ±0.16 | ±0.33 | ±0.54 | ±0.43 | ±0.39 | ±0.17 | ±0.27 |
| w/o *Rel* in $K,V$ | 36.35 | 31.42 | 33.70 | 30.05 | 22.68 | 25.83 | 31.17 | 26.11 | 28.41 | 54.08 | 50.58 | 52.27 |
| | ±0.34 | ±0.62 | ±0.31 | ±0.84 | ±1.04 | ±0.52 | ±0.85 | ±0.66 | ±0.58 | ±0.99 | ±0.72 | ±0.61 |

**Relative path encoding** For TPTrans (Eq.6 + Eq.7), We first remove the relative path encoding in the weighted sum of Value (2 in Eq.7) and then further remove the relative path in Query-Key product (1 in Eq.6), showing in the top half of Table 5. We find that with the elimination of relative path encoding gradually, the performance of TPTrans decreases obviously. After removing all submodules of relative path encoding (1 in Eq.6 and 2 in Eq.7), TPTrans totally degenerates to vanilla Transformer and performs the worst. These results highlight the usefulness of relative path encoding, both in Query-Key product (Eq.6) and the weighted sum of Value (Eq.7).

**Absolute path encoding** TPTrans-$\alpha$ extra adds absolute path encoding on Query-Key product of attention score (Eq.10) compared to TPTrans. We remove all the relative path encoding of TPTrans-$\alpha$ and compare it to vanilla Transformer shown in the last rows of both top and bottom halves of Table 5, respectively. We find that absolute path encoding significantly improves vanilla Transformer, highlighting its effectiveness.

**Relationship between them** We analyze the interaction between these two kinds of paths further. In the bottom half of Table 5, we firstly find that with the elimination of two submodules of relative path encoding gradually (1 in Eq.10 and 2 in Eq.7), the performance of TPTrans-$\alpha$ decreases at the same time. This phenomenon is consistent with the ablation study for relative path encoding shown in the previous paragraph. We then compare the top and bottom halves of Table 5 row by row from down to up and find that with the introduction of relative path gradually, the performance gap between TPTrans and TPTrans-$\alpha$ correspondingly decreases, which means the improvement brought by absolute path becomes smaller at the same time. After introducing full of the relative path encoding, the gain from absolute path encoding is almost weak. [6] These results show that these two kinds of paths are not orthogonal and much feature overlap indeed exists between them. Actually, if two sub-tokens are too *far* away from each other in the syntax tree, the relative path between them then spans the root of AST, in which the relative and absolute path encodings play the same role for modelling structure. The experimental results also show that it is preferable to encode the relative path than the absolute path.

## 5.3 Ablation study

We explore the roles of each part of our approaches. We choose TPTrans with the base setting and mainly experiment on Python and Ruby datasets.

---

[6]We also tried another method adding absolute path representation to word embedding directly in [Kim et al., 2020] as mentioned before, but the overall conclusion remains all the same.

**Path vs. distance**  To verify the benefit of learning from paths than distances presented in [Zügner et al., 2021], we try to convert all nodes in paths to the same one before feeding paths into TPTrans and present in Table 6. After such special preprocessing, what TPTrans learns completely degenerates into hops across the syntax tree between two terminals. We find that the performance of our proposed model shows noticeable declines after information ablation. It indicates TPTrans indeed learns from rich node combinations within paths, and it is better to model path than distance.

Table 6: Ablation study on path information

| Model | Python | | | Ruby | | |
|---|---|---|---|---|---|---|
| | Prec. | Rec. | F1 | Prec. | Rec. | F1 |
| TPTrans | **38.45**±0.17 | **33.63**±0.06 | **35.88**±0.05 | **32.70**±1.32 | **27.75**±0.38 | **30.02**±0.72 |
| *degenerate to hops* | 37.31±0.22 | 32.76±0.24 | 34.89±0.09 | 29.68±0.04 | 24.85±0.17 | 27.05±0.10 |

**Normalization**  As mentioned before, embedding vector sequences into the attention module of Transformer is not just trivial, so we carefully design the normalization method and present the ablation results in Table 7. These results prove that the normalization method is crucial for embedding vector sequence into attention, which is also helpful to guide the design of neural networks.

Table 7: Ablation study on normalization

| Model | Python | | | Ruby | | |
|---|---|---|---|---|---|---|
| | Prec. | Rec. | F1 | Prec. | Rec. | F1 |
| TPTrans | **38.45**±0.17 | **33.63**±0.06 | **35.88**±0.05 | **32.70**±1.32 | **27.75**±0.38 | **30.02**±0.72 |
| *w/o normalization* | 37.15±0.44 | 32.63±0.35 | 34.74±0.22 | 26.81±0.55 | 21.81±0.31 | 24.05±0.19 |

**Deeper model architecture**  We re-design a deeper model architecture for TPTrans and vanilla Transformer and set $L_E$=$L_D$=6, $D$=512, $D_{FF}$=2048 and $H$=8. For TPTrans, the input and output sizes of GRU are both 64. We present the results in Table 8 and conclude that our approaches of integrating path encodings still work for the deep model setting.

Table 8: Ablation study on deeper model

| Model | Python | | | Ruby | | |
|---|---|---|---|---|---|---|
| | Prec. | Rec. | F1 | Prec. | Rec. | F1 |
| TPTrans | **37.27**±0.36 | **33.38**±0.23 | **35.22**±0.28 | **31.10**±0.41 | **26.87**±0.74 | **28.82**±0.30 |
| Transformer | 32.71±0.27 | 27.63±0.10 | 29.96±0.16 | 24.26±0.33 | 19.66±0.45 | 21.71±0.25 |

## 6  Conclusion

In this paper, we investigate the interaction between the absolute and relative path encodings by integrating them into the unified Transformer framework, and confirm that feature overlap exists between these two kinds of paths. The detailed empirical study for path encodings also leads to our novel code representation model TPTrans. Extensive experiments and ablation study demonstrate the effectiveness of our approaches on code summarization across four different programming languages.

## 7  Acknowledgements

We thank all reviewers for their constructive comments. This research is supported by the National Key R&D Program of China under Grant No. 2020AAA0109400, and the National Natural Science Foundation of China under Grant Nos. 62072007, 61832009, 61620106007.

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
