# A Appendix

## A.1 Data preprocessing

1. Remove all the anonymous functions in datasets. Anonymous functions exist in the Javascript dataset, while we can not use these functions for code summarization task.
2. Parse a code snippet into an AST using the open-source AST parser Tree-Sitter across four different languages.
3. Traverse up from each terminal to the tree root in AST. We then get a node list for each terminal, consisting of the terminal itself and some nonterminals.
4. Depending on the literal of the single terminal for each node list, do as follows:
   1) If the literal is punctuation, then delete this node list. Punctuation tokens (such as points or brackets) do not improve code summarization performance shown in previous works.
   2) If the literal is a hardcoded string, a number or the single method name of the function, then replace the terminal with a new node whose literal is the special token `<STR>`, `<NUM>` or `<METHOD>`.
   3) If the literal can be split into sub-tokens, copy this node list multiple times and replace the terminal of each node list with a new node whose literal is each sub-token, respectively.
5. Pick up the single terminal of each node list orderly and compose the sub-token sequence as the input of Transformer.
6. For each node list, pick up all nonterminals therein and compose them as the absolute path.
7. Combine node lists in pairs and compose the relative path between two sub-tokens.

## A.2 Efficient computation for integrating path encodings

**Relative path encoding** The naive way for relative path encoding needs to encode all the pairwise paths for each code snippet and compose the relative path representation matrix $R \in \mathbb{R}^{n \times n \times d}$, where $n$ is the length of code sequence and $d$ is the dimension of the hidden representations. This approach costs $\mathcal{O}(n^2)$ for time complexity, since the number of pairwise paths is $\frac{n(n-1)}{2}$. It also needs to keep the huge matrix $R$ in GPU memory which costs $\mathcal{O}(n^2)$ for space complexity and raises the out-of-memory error for mini-batch training. To optimize them, we notice much repetition exists in all pairwise paths for a code snippet, which means we can encode the unique path only once. On this principle, we present a new method to reduce the cost. For each sample, we do as follows:

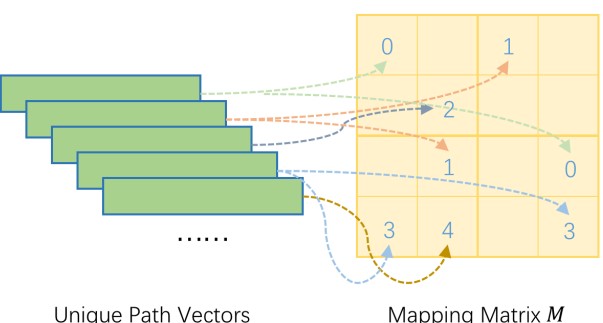

Unique Path Vectors        Mapping Matrix $M$

Figure 2: The matrix $M$ records the mapping relationship between token pairs and the relative paths for them. Specifically, $M[i,j]$ is the id of the relative path for tokens pair $(i,j)$ in all unique paths.

1. Find all the **unique** paths in all pairwise paths, and record the mapping between token pairs and the id of unique paths as matrix $M \in \mathbb{R}^{n \times n}$ (See Fig2).
2. Encode all the unique paths to vectors and compose the unique path representation matrix $R \in \mathbb{R}^{l \times d}$, where $l$ is the number of unique paths and far less than the square of code length.

3. Compute the Query-Key product $S' \in \mathbb{R}^{n \times l}$ between word representation $X \in \mathbb{R}^{n \times d}$ and the unique path representation $R$. Then **gather** the product $S'$ using matrix $M$ and get the pre-softmax attention score $S_r \in \mathbb{R}^{n \times n}$.

4. Compute the attention score $S_w \in \mathbb{R}^{n \times n}$ between words representation. Add $S_w$ to $S_r$ and compute the softmax of sum score, then get the final probability distribution of attention $A \in \mathbb{R}^{n \times n}$.

5. For each token, **scatter** the attention distribution and assign attention probability to each unique path, which means convert the matrix $A$ into $A' \in \mathbb{R}^{n \times l}$ using matrix $M$. Then compute the weighted sum of unique path representation and add it to the weighted sum of word representation.

We set the max number of unique paths 512, which covers most of the samples. The new approach shown above avoids encoding $\mathcal{O}(n^2)$ paths and reduces the time complexity into $\mathcal{O}(l)$. It reduces the size of path representation matrix $R$ from $\mathcal{O}(n^2 d)$ to $\mathcal{O}(ld)$, which ensures mini-batch training on GPUs without out-of-memory error. The mapping matrix $M$ and the gather and scatter operations introduced in the new approach do not increase the complexity compared to the naive approach.

**Absolute path encoding**     Since the number of the absolute path is $\mathcal{O}(n)$ where $n$ is the length of the code sequence, the naive way for absolute path encoding is acceptable for both time and space complexity. But we also reduce its complexity in the same way as the relative path encoding and set the max unique path number 256 for it.