# OpenReview forum: "Integrating Tree Path in Transformer for Code Representation"
_NeurIPS.cc/2021/Conference — NeurIPS 2021 Poster_

### Official Review · Reviewer_FGbv · 2021-07-14

**Rating:** 7
**Confidence:** 4

**Summary:**

This work continues on earlier advances in how positional and relational encoding is used by the self attention primitive of Transformers, in particular in the context of source-code understanding. Prior work has biased self attention to encode relative relational information between inputs (e.g., edges or distance on some graph), and to separate the self-attention term due to position from that due to input.

For the former, TPTrans runs the path between any two tokens (specifically, the path in the code Parse Tree between two leaves -- the authors call this the "relative path") through an RNN, and uses the resulting embedding to bias the self attention between those tokens.

For the latter, TPTrans-$\alpha$ runs the path from the parse-tree root to a token through an RNN (the "absolute path"), and uses that to construct a separate positional self-attention term.

The results show that both TPTrans and TPTrans-$\alpha$ outperform prior work (most notably, Code Transformer), but TPTrans seems to dominate TPTrans-$\alpha$ with a smaller set of trainable parameters.

**Ethical Concerns:**

No concerns.

**Limitations And Societal Impact:**

No concerns.

**Main Review:**

# Overall

* Originality: This is an interesting twist to prior advances (GREAT and Code Transformer, and perhaps TUPE). It's a natural twist, but a non-trivial one.

* Quality: The quality of the evaluation was mostly good, with small issues (e.g., the lack of confidence intervals). I particularly appreciated the authors showing TPTrans-$\alpha$, even though it was largely a negative result.

* Clarity: The presentation was good, especially Section 3. It gave a very precise and contextualized description of what's new and what isn't, which is greatly appreciated. The experimental section was harder to understand, and I don't have a good understanding of what precise baselines were run, and whether the comparisons are fair.

* Significance: This is still a new area, and the proposal here is likely to influence future Transformer-like ML architectures for code  or other structured inputs. As such, it's a non-trivial new contribution.


# Other Comments

I really appreciated the detailed contrast between what had been done before (Section 3.0) and what TPTrans is introducing (Section 3.1). This helps distill the differences and novel contributions, especially contextualizing them to provide motivation. In this effort, it would have been even more helpful to the reader to use the same notation where it wouldn't affect semantics. For examples, there's no reason why $a_i$ and $a_j$ in Equations 8 and 9 can't be denoted with $p_i$ and $p_j$ from Equation 3. Simialrly for $W_a^{Q/K}$ and $U^{Q/K}$.

It would be helpful to provide confidence intervals for your results. It's not clear if your improvements are statistically significant otherwise.

I found some parallels in absolute/relative paths to AnyCodeGen's evolution of code2seq (Alon, Uri, Roy Sadaka, Omer Levy, and Eran Yahav. "Structural language models of code." In International Conference on Machine Learning, pp. 245-256. PMLR, 2020.) I found it rather odd that it was not cited and mentioned.

Although removing the layer index from equations (e.g., 6, 7, ...) is helpful, I found it problematic for Equation 5. Please add a precise definition of $\mathit{Path}_{x_i\rightarrow x_j}$. Does this change for every layer, or is it dependent solely on the initial embeddings of node types?

# Smaller Issues

* Your Hellendoorn 2019 paper was at ICLR2020, not ICLR2019.

* Line 216: Table 4 should be Table 1.

* Line 221: Please explain equal interval sampling. How do you choose the interval? Is it fixed or chosen to provide a sample of length 32?

* Lines 245--249: Pointer decoders have been used in the relevant literature for different purposes. Copying (which is what you do here) or to identify positions for an edit (e.g., in HOPPITY and Hellendoorn et al 2020). Please clarify that you're using pointers for copying, not as part of the task definition. Also, please explain how you use the pointer network exactly. The information you provide does not allow the reader to reproduce your architecture.

* Lines 260--271: The text here is very hard to parse and to understand how exactly you have constructed your baselines.

* References: I found the order of citations very distracting. It was very hard to locate a citation, given that the references are provided in the order of appearance rather than alphabetical order.


**Time Spent Reviewing:**

4

---

> ### Author Response · Authors · 2021-08-10
> **Response**
>
> Thanks for your careful and valuable comments. We will explain your concerns point by point.
>
> Before the formal answers, we want to point out the minor writing error of Eq.10, which should be rewrite as
> $\alpha_{ij}=\frac{1}{\sqrt{d}}[(x_iW^Q)(x_jW^K+r_{ij}W^K_{r})^T+(a_iW^Q_{a})(a_jW^K_{a})^T]$. Sorry for any confusion due to this flaw.
>
> Q1: The main experimental results lack confidence intervals. Besides, it would be appreciated to show the result of TPTrans-alpha, as though it was largely a negative result.
>
> A1: For the concern about the main result in Table2, we repeat the experiment several times for confidence measure, and get that:
>
> |Model|Python.P|Python.R|Python.F1|Ruby.P|Ruby.R|Ruby.F1|
> |----|----|----|----|----|----|----|
> |TPTrans|38.76 $\pm$ 0.49|34.66 $\pm$ 0.35|36.59 $\pm$ 0.02|32.40 $\pm$ 0.84|27.63 $\pm$ 0.78|29.82 $\pm$ 0.79|
>
> |Model|JS.P|JS.R|JS.F1|Go.P|Go.R|Go.F1|
> |----|----|----|----|----|----|----|
> |TPTrans|33.83 $\pm$ 0.55|28.37 $\pm$ 0.54|30.86 $\pm$ 0.53|55.79 $\pm$ 1.3|51.05 $\pm$ 1.3|53.30 $\pm$ 0.83|
>
> We further compare the improvement of the SOTA(Code Transformer) against the previous model(Here we pick up XLNet for comparison), and the gain of our proposed model over Code Transformer.
>
> The gain of Code Transformer over XLnet:
>
> |Model|Python.F1|Ruby.F1|JS.F1|Go.F1|
> |----|----|----|----|----|
> |CT minus XLNet|0.5|-0.19|2.38|1.73|
>
> And the gain of our model over Code Transformer:
>
> |Model|Python.F1|Ruby.F1|JS.F1|Go.F1|
> |----|----|----|----|----|
> |TPTrans minus CT|1.6|2.27|-1.27|1.97|
>
> The code transformer is designed based XLNet and enhanced with multiply structural relations of code AST.
> We can then find that in most datasets, compared the improvement of code transformer over XLNet, our model obtains more remarkable improvement over code transformer, although Code Transformer is well designed for code and XLNet is not.
>
>
> For the concern about TPTrans-alpha, we have shown its performance in Section 5.2 (Ablation Study, L300-L305) compared to TPTrans. Please note that unlike the models presented in Table2, the models presented in Section 5.2 are all not equipped with pointer networks. But we still are glad to show the performance of TPTrans-alpha equipped with the pointer(We are so sorry for only showing the results of Ruby and Js datasets due to tight time. And we will try our best to complete the results of the other two datasets getting enough time.), that is:
>
> |Model|Ruby.P|Ruby.R|Ruby.F1|JS.P|JS.R|JS.F1|
> |----|----|----|----|----|----|----|
> |TPTrans|32.40 $\pm$ 0.84|27.63 $\pm$ 0.78|29.82 $\pm$ 0.79|33.83 $\pm$ 0.55|28.37 $\pm$ 0.54|30.86 $\pm$ 0.53|
> |TPTrans-alpha|33.07 $\pm$ 0.86|28.34 $\pm$ 0.61|30.52 $\pm$ 0.53|33.68 $\pm$ 0.34|28.95 $\pm$ 0.54|31.14 $\pm$ 0.46|
>
> Here, we want to make more explanation about the relationship between relative path and absolute path. From Table3 presented in Section 5.2(Ablation Study), we conclude that after introducing full of the relative path encoding, the improvement from the absolute path encoding almost diminishes. We feel much sorry for that, because of lacking error bars for Table3, readers maybe deduce a misguided conclusion that a concrete conflict definitely exists between two kinds of paths. On the contrary, actually, the only thing we firmly conclude here is that these two kinds of path encoding methods are not orthogonal, which means overlap exists rather than conflict. From the above supplemental results of TPTrans and TPTrans-alpha both equipped with pointers, we conclude that although the improvement of TPTrans-alpha over TPTrans is not consistently significant, the so-called 'conflict is also not a certainty.
>
> We feel much sorry for any confusion. We will fix the misleading and even somewhat exaggerated explanation at L315-L319 and update the experiment results with error bars for Table3 in the later version.
>
>
>
> Q2: The experimental section was harder to understand, and it is hard to have a good understanding of what precise baselines were run, and whether the comparisons are fair.
>
> A2: We are much sorry for missing details in this section. The baselines we choose are Code2Seq, Great, XLNet and Code Transformer.
>
> The code2seq is a non-transformer model and use pairwise path information in AST for code representation. The Great is a Transformer-based model, and it additionally biases the attention vias manually designed structural edges. Code Transformer is another Transformer-based model, incorporating multiple different relations to leverage code structure. Both Code2seq, Great and Code Transformer are well designed for learning code representation.
>
> Besides, we also pick the XLNet as one of the baselines. It is important to note that the XLNet is not designed for learning source code representation, and it is designed initially for huge-corpus natural language pretraining in [1]. We list XLNet for comparison because the Code Transformer is designed based on XLNet and additionally incorporates multiply structural relations within code syntax trees over it. Although the XLnet is not specially designed for source code, its novel architecture of two-stream self-attention still shows powerful performance in code representation learning. For more details about XLnet and Code Transformer, please refer to [1] and [2], respectively.
>
> In the main result, all models (all baselines and our model) are trained from random initial parameters. For fair comparisons, we set 1024 hidden dims and 3 layers for Transformer, which keeps a similar setting with the backbone of Code Transformer present in [2]. The additional parameters we introduce for the encoding path come from two parts: the lookup table embedding matrix for nodes and two 1-layer bidirectional GRUs(one for relative path and another for absolute path).
>
> 1) The dimension of the embedding matrix of nodes is 64, while for the Python language, the vocab size for nodes is 105 (105 for Ruby and Js, 94 for Go). So the num of parameters is 6720 for Python (6720 for Ruby and JS, 6016 for Go).
>
> 2) The input and output dim of GRUS are 64 and 64*2( *2 for bi-direction), respectively, so the total parameters is 49920.
>
> The total parameters of the vanilla Transformer are about 100M (including the parameters of embedding matrix for word's lookup table). So we can find that the proportion of additional parameters for integrating tree path is only about 0.057%. This shows that the improvement of our methods mainly comes from good design instead of parameters' increase.
>
>
> Q3: In this effort, it would have been even more helpful to the reader to use the same notation where it wouldn't affect semantics. For examples, there's no reason why $a_i$ and $a_j$ in Equations 8 and 9 can't be denoted with $p_i$ and $p_j$ from Equation 3. Similarly for $W_{a}^{Q/K}$ and $U^{Q/K}$.
>
>
> A3: Thanks for your kind advice. Actually, the notations in the Eq.5-10 are not designed arbitrarily.
>
> 1) In Eq.5-7, we use the notation $r_{ij}$ to represent the vector of the relative path between token $i$ and $j$. The $r$ is the first letter of the word 'relative'. For the same reason, the projection matrices of Key and Value $W^K_{r}$ and $W^Q_{r}$ in Eq.6 and Eq.7 also add sub $r$ to represent the 'relative'.
>
> 2) In Eq.8-10, we use notation $a_i$ to represent the absolute path vector for token $i$. Similarly, the $a$ is the first letter of 'absolute', and we also add sub $a$ for matrice $W^Q_{a}$ and $W^K_{a}$ to represent the 'absolute'.
>
> Q4: I found some parallels in absolute/relative paths to AnyCodeGen's evolution of code2seq (Alon, Uri, Roy Sadaka, Omer Levy, and Eran Yahav. "Structural language models of code." In International Conference on Machine Learning, pp. 245-256. PMLR, 2020.) I found it rather odd that it was not cited and mentioned.
>
> A4: Thanks so much for pointing out this issue. We did notice this paper during our research, but we forgot to cite it due to our writing oversight. We will carefully fix it in the later version. Thanks again for the kind reminder.
>
>
> Q5: Although removing the layer index from equations is helpful, I found it problematic for Equation 5. Please add a precise definition of. Does this change for every layer, or is it dependent solely on the initial embeddings of node types?
>
> A5: We are much sorry for the confusion due to missing details. The $Path(x_i\rightarrow x_j)$ in Eq.5 is a vector list consisted with [$n_0$,$n_1$,$n_2$,...,$n_m$], where $m$ is the path length and each $n$ is the embedding vector for each node which is looked up from the embedding matrix table of node type. We then use a bi-directional GRU to encode the input sequence and get the final state of the path. After that, we share this path representation across different layers and heads of the Transformer for efficiency. In other words, we pre-encode the path by GRU before feeding features into Transformer, and we encode each path only one time using only one GRU.

---

> > ### Comment · Reviewer_FGbv · 2021-08-24
> > **After "Response" and "Response 2"**
> >
> > Thank you for the very detailed responses to my questions. They were very helpful and my remaining concerns are addressed.
> >
> > It seems that given the direct comparison of TPTrans and TPTrans-$\alpha$ with confidence intervals in the response, it's unclear that TPTrans-$\alpha$ has any cases in which it dominates. Is this true? Either way, it's a valuable design point and useful to present, so thank you for including it.

---

> > > ### Author Response · Authors · 2021-08-24
> > > **Response**
> > >
> > > Yes. From our experimental result, we conclude that much performance overlap exists between the two paths. That means no concrete evidence shows TPTrans-$\alpha$ dominate over TPTrans, although the former shows better in some datasets.

---

> ### Author Response · Authors · 2021-08-10
> **Response2**
>
> Q6: Please explain equal interval sampling. How do you choose the interval? Is it fixed or chosen to provide a sample of length 32?
>
> A6: Actually, the relative path may be short (for example, one node exists in the path if two terminals have the same parent), but can also be long, and the max length is equal to twice the max absolute path length (for example, if two terminals are too far away from each other in the syntax tree, the relative path between them maybe span the root of a tree). To handle that, we set the max path length as 32, which covers almost all of the relative paths. For a few paths whose length exceeds 32, we adopt equal interval sampling to retain the information as much as possible.
>
> When the input data length exceeds the max, the usual approach in the NLP community is to truncate the last tokens. However, since the relative path represents the relationship between two tokens, the node at both the left and right sides of the path are equally important. For example, the leftmost and rightmost nodes in the relative path are the parents of these two terminals, respectively. Therefore, we adopt equal interval sampling instead of truncating last, and the pseudocode is:
>
>     'if len(path) > args.max_path_length:
>         idx = np.linspace(0, len(path) - 1, args.max_path_length, dtype=int).tolist()'.
>
> The sampling interval is not fixed, and it depends on the real length of the path. After that, we index nodes from the path and reassemble them.
>
> Finally, we also want to emphasize again that the max path length 32 has already covered almost the relative paths, so the interval sampling can also be considered as the icing on the cake.
>
> Q7: Please clarify that you're using pointers for copying, not as part of the task definition. Also, please explain how you use the pointer network exactly. The information you provide does not allow the reader to reproduce your architecture.
>
> A7: Much sorry for the missing details for the pointer network. Overall, we refer to the source code of Code Transformer released in [2] and add the same pointer network on our model for fair comparisons. The pointer network enhances predictions by pointing at positions and is often presented in many seq2seq models. Concretely, both Code Transformer and our model use the pointer network equipped with the sentinel[3]. At each decoding step, the decoder additionally calculates a dot-product between the target sequence features and the sentinel (presented as learnable vector parameters) and takes the product as the gated probability of whether to copy, which is slightly different from the pointer network present in [4]. We will release our source code for result reproduction and more details.
>
> For other smaller issues:
> 1) The citation for Hellendoorn 2019;
> 2) Line 216: Table 4 should be Table 1;
> 3) Reference: the order of citations is distracting;
>
> we appreciate your kind reminder, and we will carefully fix them in the later version.
>
> [1]XLNet: Generalized Autoregressive Pretraining for Language Understanding
>
> [2]Language-Agnostic Representation Learning of Source Code from Structure and Context
>
> [3]Pointer sentinel mixture models
>
> [4]Pointer networks

---

### Official Review · Reviewer_SWC5 · 2021-07-15

**Rating:** 4
**Confidence:** 3

**Summary:**

This paper introduces a new neural network architecture for code, TPTrans (Tree Path Transformer), by modifying the attention calculation in the Transformer architecture to use information about paths in the syntax trees of programs. The modification works by computing path embeddings between pairs of leaf nodes ("relative path embeddings") and between leaf nodes and the root (forming nodes' "absolute path embeddings"), and incorporating these into the Transformer's attention module. The paper evaluates TPTrans, including baselines and variants, on code summarization -- the task of predicting a function's name given its body -- for four different programming languages. The main experiments reveal the value of including embeddings of sequences (the paths) in the attention calculation over baselines, and subsequent experimentation compares the impact of relative and absolute paths, and the effect of pointer network outputs on the model.

**Ethical Concerns:**

Parts of the introduction to Section 3 are lifted from Section 3 of [Language-Agnostic Representation Learning of Source Code from Structure and Context](https://arxiv.org/abs/2103.11318).

**Ethics Review Area:**

["Research Integrity Issues (e.g., plagiarism)"]

**Limitations And Societal Impact:**

A technical limitation of the work is: The method relies on having a parse for the program, which limits the approach to well formed code and languages supported by tree-sitter. The baseline Transformer does not have this limitation. The paper does not address potential societal impact.

**Main Review:**

The paper's core contribution is the introduction of TPTrans, a method of incorporating paths from a syntax tree into the attention module of the Transformer architecture. By evaluating this approach with two kinds of paths on code summarization, and by finding it outperforms baseline methods, the paper indeed shows the feasibility of incorporating sequential information into transformer attention as an inductive bias, as claimed.

This is an important area of study, as machine learning for source code is an ever increasingly important domain, yet techniques from natural language processing such as the transformer architecture do not take advantage of properties of source code. Identifying models that learn better representations of source code and fully take advantage of the information and structure present in source code, including its semantics, is a critical challenge. This paper is a step toward this, identifying a method of incorporating program structure for better learned representations of code.

The method proposed for incorporating relative and absolute path encodings into Transformer attention is conceptually simple, and is explained clearly in Section 3.1. The conceptual simplicity of the proposal is a strength of the paper, as is the clarity of the description in this section. The main result, showing on most tasks that the relative path variant TPTrans outperform all baselines including the absolute path variant TPTrans-$\alpha$, is another strength.

The specific choices of absolute path for each node and relative path for each node-pair makes the method specific to code. These paths are embedded using a GRU over the sequence representation of the path. Beyond these choices, the approach is more general, applicable for incorporating any pairwise and element-wise node information about a sequence

Considered individually these components are not novel. GRUs are a standard way of encoding sequential information, the transformer is a standard sequence-processing architecture, the use of absolute position encodings is present in the original transformer and relative position encodings are introduced in [1]. The paper's strength is in it's combination of these components, demonstrating that sequence information is suitable for both relative and absolute position encodings, and that in the context of source code choosing these position encodings improves performance on an important benchmark task. That said, since TPTrans is primarily a novel combination of existing architecture components and an extension of the transformer attention module similar to ways it has already been extended, the size and significance of the contribution is limited.

Evaluations are done on the task of code summarization, which is a sensible task for evaluating performance on representation learning of source code. Four programming languages are considered, Python, Javascript, Go, and Ruby, and TPTrans outperforms the baselines considered on all languages but Javascript. One area for improvement in the experimental setup is clarifying the axes along which the models are comparable. Do they have comparable parameter counts? If the increase in performance were only due to an increase in parameter count, which could also be achieved by scaling up existing models, that would decrease the significance of the result.

A weakness of the experiments is they lack error bars or any measure of confidence or variation. Independent of this, the experimental evidence is also somewhat thin, as a single task is used to demonstrate improvements on the quite broad goal of representation learning.

One potential way to improve the paper would be to demonstrate why the path embeddings are useful. For example, the model admits inspecting which path embeddings are attended to. Do these exhibit any patterns or have any particular semantic meaning?

As an aside, the paper unfortunately contains many grammatical and typographic errors. Usually these do not impede clarity too much, and in my review score I do not count them against the paper. Some of them do make the paper more difficult to read, however (e.g. lines 269-270), but in all cases I think I was able to determine the intent of the sentence.

[1] Self-attention with relative position representations.

**Needs Ethics Review:**

Yes

**Time Spent Reviewing:**

8

---

> ### Author Response · Authors · 2021-08-10
> **Response**
>
> Thanks for your careful and valuable comments. We will explain your concerns point by point.
>
> Before the formal answers, we want to point out the minor writing error of Eq.10, which should be rewrite as
> $\alpha_{ij}=\frac{1}{\sqrt{d}}[(x_iW^Q)(x_jW^K+r_{ij}W^K_{r})^T+(a_iW^Q_{a})(a_jW^K_{a})^T]$. Sorry for any confusion due to this flaw.
>
> Q1: One area for improvement in the experimental setup is clarifying the axes along which the models are comparable. Do they (baselines and proposed models) have comparable parameter counts? If the increase in performance were only due to an increase in parameter count, which could also be achieved by scaling up existing models, that would decrease the significance of the result.
>
> A1: Thanks for your concern. For fair comparisons, we set 1024 hidden dims and 3 layers for Transformer, which keeps a similar setting with the backbone of Code Transformer present in [1]. The additional parameters we introduce for the encoding path come from two parts: the lookup table embedding matrix for nodes and two 1-layer bidirectional GRUs(one for relative path and another for absolute path).
>
> 1) The dimension of the embedding matrix of nodes is 64, while for the Python language, the vocab size for nodes is 105 (105 for Ruby and Js, 94 for Go). So the num of parameters is 6720 for Python (6720 for Ruby and JS, 6016 for Go).
>
> 2) The input and output dim of GRUS are 64 and 64*2( *2 for bi-direction), respectively, so the total parameters is 49920.
>
> The total parameters of the vanilla Transformer are about 100M (including the parameters of embedding matrix for word's lookup table). So we can find that the proportion of additional parameters for integrating tree path is only about 0.057%. This shows that the improvement of our methods mainly comes from good design instead of parameters' increase.
>
> Q2: A weakness of the experiments is they lack error bars or any measure of confidence or variation. Independent of this, the experimental evidence is also somewhat thin, as a single task is used to demonstrate improvements on the quite broad goal of representation learning.
>
> A2: For the concern about the main result in Table2, we repeat the experiment several times for confidence measure, and get that:
>
> |Model|Python.P|Python.R|Python.F1|Ruby.P|Ruby.R|Ruby.F1|
> |----|----|----|----|----|----|----|
> |TPTrans|38.76 $\pm$ 0.49|34.66 $\pm$ 0.35|36.59 $\pm$ 0.02|32.40 $\pm$ 0.84|27.63 $\pm$ 0.78|29.82 $\pm$ 0.79|
>
> |Model|JS.P|JS.R|JS.F1|Go.P|Go.R|Go.F1|
> |----|----|----|----|----|----|----|
> |TPTrans|33.83 $\pm$ 0.55|28.37 $\pm$ 0.54|30.86 $\pm$ 0.53|55.79 $\pm$ 1.3|51.05 $\pm$ 1.3|53.30 $\pm$ 0.83|
>
> We further compare the improvement of the SOTA(Code Transformer) against the previous model(Here we pick up XLNet for comparison),
> and the gain of our proposed model over Code Transformer:
>
> The gain of Code Transformer over XLnet:
>
> |Model|Python.F1|Ruby.F1|JS.F1|Go.F1|
> |----|----|----|----|----|
> |CT minus XLNet|0.5|-0.19|2.38|1.73|
>
> And the gain of our model over Code Transformer:
>
> |Model|Python.F1|Ruby.F1|JS.F1|Go.F1|
> |----|----|----|----|----|
> |TPTrans minus CT|1.6|2.27|-1.27|1.97|
>
> The code transformer is designed based XLNet and enhanced with multiply structural relations of code AST.
> We can find that in most datasets, compared to the improvement of code transformer over XLNet, our model obtains more remarkable improvement over code transformer, although Code Transformer is well designed for code and XLNet is not.
>
> We very much appreciate your criticism of the experimental task setting. As we all know, the method naming task is a popular and useful benchmark for code representation learning. Many works pick this task to show the ability of code representation learning[1][2]. In our paper, we experiment on four datasets across different languages to show our approaches' effectiveness. We're indeed happy to explore other tasks such as classification and retrieval and also have confidence in our proposed model.
>
> Q3: One potential way to improve the paper would be to demonstrate why the path embeddings are useful. For example, the model admits inspecting which path embeddings are attended to. Do these exhibit any patterns or have any particular semantic meaning?
>
> A3: Thanks for your concern. First of all, many studies have shown that encoding syntax tree paths are efficient for code representation learning[2-4], and our experiments also draw similar conclusions. As summarized in our paper (L66-L70), the relative path can reflect the structural relationship between two tokens and the absolute path can complement the structural information for each token and reflect the program behaviour on it.
>
> As you concern, specific paths should receive more attention than others given one code snippet. For this reason, the Code2seq[2] equipped an attention module for the sequence decoder. In this paper, we modify the formula of relative position encoding (Eq.4, presented in [6]) to integrate the relative path encoding into Transformer (Eq.6 and Eq.7) and extract valuable information from paths in a data-driven manner. In [6], the authors say: 'The edge between input elements $x_i$ and $x_j$ is represented by vectors $a_{ij}^{V}$ and $a_{ij}^{K}$.'  That means that the pairwise relationship between two tokens is considered as an edge, and these two tokens can be considered as two nodes at head and tail, respectively. So, both the motivations of Eq.4 and Eq.6+Eq.7 are to establish the connection between $Vec(Head)$ and the $Vec(Tail)+Vec(Edge)$, which means that the attention module of TPTrans also focuses on the path information between two tokens. As a result, each token has the potential to assign big attention scores for other tokens, which connect it with more representative paths. Besides, after fully integrating the tree path, the transformer's decoder can also have the potential to assign higher attention scores for the terminals attached to important paths.
>
> Q4: The method relies on having a parse for the program, which limits the approach to well formed code and languages supported by tree-sitter. The baseline Transformer does not have this limitation
>
> A4: To learn structured code representation, we always need to parse the code snippet first. Our primary baseline, Code Transformer, chooses the Semantic library as the parser for Python, Js, Ruby and Go languages (see appendix of [1] for more details). In this paper, we choose the open-source tree-sitter as the code parser. So we firmly believe the view, we have the limitation of parser while the baseline Transformer does not have, is not reasonable.
>
>
> Q5: Parts of the introduction to Section 3 are lifted from Section 3 of Language-Agnostic Representation Learning of Source Code from Structure and Context.
>
> A5: Thanks so much for pointing out our writing oversight. We re-examine Section 3 and indeed find that some sentences are quite similar to existing papers. We will carefully fix them in the later version. Thanks again for the kind reminder.
>
>
> [1]Language-Agnostic Representation Learning of Source Code from Structure and Context
>
> [2]Code2seq: Generating Sequences from Structured Representations of Code
>
> [3]TreeBERT: A Tree-Based Pre-Trained Model for Programming Language
>
> [4]Code prediction by feeding trees to transformers
>
> [5]Global Relational Models of Source Code
>
> [6]Self-Attention with Relative Position Representations

---

> > ### Comment · Reviewer_SWC5 · 2021-08-24
> > **Thank you for your response.**
> >
> > Thank you for your response.
> >
> > The Eq. 10 correction is noted. The parameter counts and error bars are appreciated. (What is the measure of spread in your response above?)
> >
> > Regarding parser usage, you are correct that the Code Transformer uses a parser. Of course, other Transformer baselines you could have considered would not have this limitation, but as you correctly note, the SOTA one that you chose (Code Transformer) does share this limitation with your approach.

---

> > > ### Author Response · Authors · 2021-08-26
> > > **Response**
> > >
> > > Thanks for your concerns.
> > >
> > > The result presented is Mean $\pm$ Sample Standard Deviation.
> > >
> > > As we have mentioned, the CodeTransfromer uses the Semantic Library as the multi-language parser.
> > >
> > > Another Transformer-based baseline, GREAT, uses a monolingual Python parser to extract syntax trees in its original paper[5](since only the Python dataset Py150 has been evaluated in the paper). But unfortunately, since the specialized preprocess used by GREAT is not public and also not available for the multi-language scene, the authors of CodeTransformer produce the results for GREAT use the Semantic Library as code parsers(see [1] for more details). Anyway, we can say that GREAT always needs a syntax parser to learn code _Structure_.
> > >
> > > The other Baseline XLNet does not need parsers for learning code representation since it only focuses on code _Context_ while not _Structure_.
> > >
> > > For the overall comparison of all models, please refer to the comment we released: 'One of the concerns about the difference between our model and the baselines.'.
> > > On the whole, we firmly think that code parsing is essential to learning code _Structure_.
> > >
> > > Please point out any other possible confusing statement in this paper, and we will try our best to eliminate it.

---

> > > ### Author Response · Authors · 2021-09-02
> > > **Further Comments**
> > >
> > > Thanks for your reading. And also much appreciate your valuable suggestions again.
> > >
> > > We have added many new experiments and post public comments about these critical issues. Please refer to them and point out any other possible confusing statement further. We will try our best to eliminate it.
> > >
> > > Thanks so much again.

---

### Official Review · Reviewer_43xz · 2021-07-19

**Rating:** 4
**Confidence:** 4

**Summary:**

The paper presents a way to incorporate the tree-structure information into a transformer architecture to better learn representation for code. The tree structures are first dispatched as leaf-to-leaf "relative paths" and leaf-to-root "absolute paths", and then embedded and modulated as additional learnable parts of the query, key, and value components within the transformer attention. The authors evaluate their proposal on the task of extreme code summarization -- predicting function names. Results show improvements over 4 previous non-transformer and transformer models on 3 out of 4 programming languages tested. The authors also carry out ablation studies to understand the effect of absolute paths and pointer networks but many important questions remain unanswered.

**Limitations And Societal Impact:**

The paper could benefit from a dedicated discussion about limitations.

**Main Review:**

The writing is clear and easy to follow. The paper follows the current trend of using transformers to learn code representation and incorporating richer structures into the originally sequence-based transformer model.

The proposed way for adding relative paths and absolute paths into the transformer attention seems natural but the exact places and terms added are somewhat arbitrary. For example, why the relative-path terms are added at the K part (Eq. 6) and the V part (Eq. 7) but not the Q part? For absolute paths, why the responding terms are only by themselves (Eq. 9) but not with any of the Q, K, V parts?

The paper also needs a clearer description of experimental settings, especially training settings (see Questions below), to fully make sense of the results.

The main results (Table 1) show a majority but not definitive gain. Even for where it gains, the gains are modest.

Additionally, why JS is such an outlier? Besides the under-supported claim that "JS's structure is weak", are there deeper reasons related to the dataset and the proposed model? It would be great if the authors could shed more light on the case.

I appreciate the effort the authors put into the ablation study. However, it also raises questions. Why is adding absolute paths -- more information for the model to rely on, degrades the metrics? Shall we try to add more relative-path terms since the trend in Table 3 suggests more is better? Rather than conclude that absolute paths do not really help, shall we try other ways of adding the terms and/or adding the terms to different places? Table 4 and 5 generally shows the proposal still works well in combination with sequence bias and pointer network, which is good.

In general, I see a solid effort and a sensible way to incorporating tree structures into transformers for learning code representation. However, the exact ways of incorporating paths lack justification, and the improvements in numbers are limited and the unclarity of the setting further decreases the credibility. The significance of the paper could be improved by giving more supported insight into how the tree structure, in the way it is used, is helping or not helping with learning code representation under different scenarios.

### Pros
- The writing is clear and easy to follow.
- Many potential audiences in learning code representations using structures.
- Natural way of adding tree structures as paths into transformer attention.
- Experiments show the proposal generally improves metrics.

### Cons
- The exact way of modifying attention seems arbitrary and lacks justification from either intuition or ablation study.
- Experimental results show a majority but not definitive gain, and the gains are modest. The no-gain case (JavaScript) is not well-explained.
- The efficacy of the proposed architecture is only evaluated over one training dataset and one task.
- Not too much insight was generated about how and why it works or not for future research.
- The title says "for Code Representation" but the authors only experimented on code summarization.

### Questions
- Justifications for the seemingly arbitrary places to add relative- and absolute-path terms in Eq. 5-10.

- Main results in Table 2. What are all the data used to train each model? For transformer-based baselines, did the authors train them from scratch, or used a pre-trained model and then fine-tuned over the CSN datasets? This question is related to fair comparison as well as if the models with a large number of parameters will become under-trained or overfitting to a relatively small number of training instances.

- Table 2, Table 5. Why is the trend so different for JavaScript? Besides the under-supported claim that "JS's structure is weak", could it be because the dataset has biases or unusual properties, or the proposed model itself has some "flaws"?

- L320 "If two sub-tokens are too far away ... the relative and absolute path encoding play the same role ..." Can we find more evidence to justify or falsify this? Local pairs can still benefit from absolute paths. Additionally, if there is any trimming of the paths (for example, due to input-length constraints), then absolute paths can keep information that would be otherwise discarded from (the longer) relative paths.

### Other points
- L278 "... since the structure of Javascript is much weak, ..."
- L339 "... the structure of Javascript is weaker than other languages, ..." I don't find references or supporting evidence for this claim. What makes the authors conclude that JS is less structured?

**Time Spent Reviewing:**

3

---

> ### Author Response · Authors · 2021-08-10
> **Response**
>
> Thanks for your careful and valuable comments. We will explain your concerns point by point.
>
> Before the formal answers, we want to point out the minor writing error of Eq.10, which should be rewrite as
> $\alpha_{ij}=\frac{1}{\sqrt{d}}[(x_iW^Q)(x_jW^K+r_{ij}W^K_{r})^T+(a_iW^Q_{a})(a_jW^K_{a})^T]$. Sorry for any confusion due to this flaw.
>
> Q1: The proposed way for adding relative paths and absolute paths into the transformer attention seems natural, but the exact places and terms added are somewhat arbitrary. Justifications for the seemingly arbitrary places to add relative- and absolute-path terms in Eq. 5-10 are concerned.
>
> A1: Overall, our ways of integrating tree paths are almost inspired by the positional encoding methods about Transformer in NLP. The encoding methods, not only for the position but also our proposed path encoding, are both not designed arbitrarily. We will explain our design point by point, from positional encoding and analogy to path encoding.
>
> 1) Firstly, as you can see, we modify the formula of relative position encoding (Eq.4, presented in [1]) to integrate the relative path encoding into Transformer (Eq.6 and Eq.7). In [1], the authors say: 'The edge between input elements $x_i$ and $x_j$ is represented by vectors $a_{ij}^{V}$ and $a_{ij}^{K}$.'  That means that the pairwise relationship between two tokens is considered as an edge, and these two tokens can be considered as two nodes at head and tail, respectively. So, the motivation of Eq.4 is to establish the connection between $Vec(Head)$ and the $Vec(Tail)+Vec(Edge)$. Similar ideas are also presented in other areas, such as Knowledge Graph[2]. The Translation principle in Knowledge Graph says 'the tail entity should be close to the embedding of the head entity plus vector depends on the relationship'[3].
>
>     In our paper, TPTrans encodes the pairwise path between source code tokens across the syntax tree as the edge relationship between them. So it makes sense to add the relative path at the K and V (Eq6 and Eq7), but not at the Q part.
>
> 2) Secondly, we modify the formula of absolute path encoding in TUPE (Eq.3, presented in [4]), to integrate the absolute path encoding (Eq.9). As mentioned in our paper, before the TUPE, the original Transformer present a simple way for positional encoding, in which a real-valued vector $p_i$ is added to the word embedding $w_i$ at position $i$. In this way, the attention layer can be writed as:
> $\alpha = ((w_i+p_i)W^Q)((w_j+p_j)W^K)^T$, and it can be expansion and rewrite as:
> $\alpha = (w_i W^Q)(w_j W^K)^T + (w_i W^Q)(p_j W^K)^T + (p_i W^Q)(w_j W^K)^T + (p_i W^Q)(p_j W^K)^T$.
> The TUPE's authors think it is unnecessary to model the correlations between absolute position and words, so they remove the 2nd and 3rd terms in the expansion formula. They also replace different project matrices for position and word. In this way, the TUPE unites the correlations between them and rewrite as:
> $\alpha = (w_i W^Q)(w_j W^K)^T + (p_i U^Q)(p_j U^K)^T$. The experiment results in [4] finally prove the efficiency of it (we omit the scaling term $\sqrt{d}$ for simplicity. For more details, please refer to the original paper[4]).
>
>     As for our method, we draw inspiration from the advanced positional encoding method in NLP and modify it to integrate the root-leaf path encoding (called the absolute path in our paper). The root-leaf path information reveals the code behaviour for each source code token. So we think our design makes sense, which unties the correlations for absolute path and word, similarly to TUPE. Actually, during our research, we also tried adding path representation with token embedding directly but got no improvement over our proposed method. This simple method is equivalent to embedding path information into all QKV parts, which can be an analogy to the original positional encoding method in the Transformer paper[6]. We think the phenomenon of no improvement is much similar to the conclusion of untying correlations in TUPE. After overall consideration, we believe it is reasonable for our already proposed methods.
>
>
> Q2: Main results in Table 2 are concerned. What are all the data used to train each model? For transformer-based baselines, did the authors train them from scratch, or used a pre-trained model and then fine-tuned over the CSN datasets?
>
> A2: We are much sorry for missing details in this section. In the main result, all models (all baselines and our model) are trained from random initial parameters.
> It is important to note that the XLNet is not designed for learning source code representation, and it is designed initially for huge-corpus natural language pretraining in [6]. The reason we list XLNet for comparison is that the Code Transformer is designed based on XLNet. Code Transformer additionally incorporates multiply structural relations within code syntax trees over XLNet. Although the XLnet is not specially designed for source code, its novel architecture of two-stream self-attention still shows powerful performance in code representation learning. For more details about XLnet and Code Transformer, please refer to [6] and [7], respectively.
>
> Q3: The main results show a majority but not definitive gain and the gains are modest.
>
> A3: For the concern about the main result in Table2, we repeat the experiment several times for confidence measure, and get that:
>
> |Model|Python.P|Python.R|Python.F1|Ruby.P|Ruby.R|Ruby.F1|
> |----|----|----|----|----|----|----|
> |TPTrans|38.76 $\pm$ 0.49|34.66 $\pm$ 0.35|36.59 $\pm$ 0.02|32.40 $\pm$ 0.84|27.63 $\pm$ 0.78|29.82 $\pm$ 0.79|
>
> |Model|JS.P|JS.R|JS.F1|Go.P|Go.R|Go.F1|
> |----|----|----|----|----|----|----|
> |TPTrans|33.83 $\pm$ 0.55|28.37 $\pm$ 0.54|30.86 $\pm$ 0.53|55.79 $\pm$ 1.3|51.05 $\pm$ 1.3|53.30 $\pm$ 0.83|
>
> We further compare the improvement of the SOTA(Code Transformer) against the previous model(Here we pick up XLNet for comparison),
> and the gain of our proposed model over Code Transformer.
>
> The gain of Code Transformer over XLnet:
>
> |Model|Python.F1|Ruby.F1|JS.F1|Go.F1|
> |----|----|----|----|----|
> |CT minus XLNet|0.5|-0.19|2.38|1.73|
>
> And the gain of our model over Code Transformer:
>
> |Model|Python.F1|Ruby.F1|JS.F1|Go.F1|
> |----|----|----|----|----|
> |TPTrans minus CT|1.6|2.27|-1.27|1.97|
>
> As mentioned in A2, the code transformer is designed based XLNet and enhanced with multiply structural relations of code AST.
> We can find that in most datasets, compared the improvement of code transformer over XLNet, our model obtains more remarkable improvement over code transformer, although Code Transformer is well designed for code and XLNet is not.
>
> Q4: More explanation for the ourlier of JavaScript dataset. Are there deeper reasons related to the Javascript dataset and the proposed model?
>
> A4: We appreciate your concern about the exception of the Javascript dataset.
> From Table2, we can find that the benefit of encoding relative path for Javascript dataset is not as powerful as the performances in other datasets (while the improvement over vanilla Transformer is still significant, see Table3).
> To investigate this phenomenon, we try to count the mean and variance of the length of all relative paths across four different datasets and get that:
>
> | Python | Mean | Var |  Ruby | Mean | Var |JS | Mean | Var| Go | Mean | Var|
> | ----- | ----- | ----- | ----- | ----- | -----|----- | ----- | -----| ----- | ----- | ----- |
> | Train |11.96 | 24.61| Train |9.29 | 17.62|Train |14.59 |41.46| Train |11.18 | 20.16|
> | Valid |11.81 |23.64| Valid |9.76 |18.84 | Valid |14.40| 40.02| Valid |11.56 | 22.40 |
> | Test |12.05| 25.36| Test |9.60 |18.82| Test |14.57| 40.59| Test |10.68| 18.83|
>
> In this table, we can find that the path length of Javascript is not only longer but also has a significantly bigger variance than other datasets. So we assume that due to such property of the JS dataset, it is harder to extract useful structural information from the path than other datasets. In addition, we are also surprised to find that the length and variance of paths of the Ruby dataset are smaller than other datasets. Meanwhile, the gain of TPTrans over baselines on Ruby datasets are more significant than others(please refer to A3 for details).
>
> But unfortunately, as to the outlier of path length and variance of JS languages, we have not yet reached an accurate conclusion.
> One possible line of reasoning is that the inherent property of Js language results in a much complicated syntax tree.
> We also suspect that this is due to the internal design of the language parser, and perhaps a better parser might alleviate this problem.
>
> Q5: Shall we try to add more relative-path terms since the trend in Table 3 suggests more is better?
>
> A5: From Table3, we indeed find that more relative path terms are better. But as we mentioned in the A1, only the design of relative path encoding on the K, V parts does make sense because these equations (Eq6 and Eq7) formulate the relationship among the head, tail and the edge between them. If we added path encoding both at Q and K, then that means $Vec(Tail)+Vec(Edge)$ will attend to $Vec(Head)+Vec(Edge)$, which is not reasonable.

---

> ### Author Response · Authors · 2021-08-10
> **Response2**
>
> Q6: Rather than conclude that absolute paths do not really help, shall we try other ways of adding the terms and/or adding the terms to different places?
> Why is adding absolute paths degrades the metrics?
>
> A6: Firstly, it's worth emphasizing that the enhanced absolute path encoding over the vanilla transformer is much efficient, which can be concluded by comparing the result of the last rows of both top and bottom halves at Table3 (described at L300-L305). But overall, if we integrate into Transformer both relative and absolute path encoding, the improvement becomes to diminish compared to the only encoding relative path. Actually, we have tried different choices to add encoding terms during our research, as your concern. But as we mentioned at A1, not all combinations of adding terms make sense and are reasonable. The only alternative that makes sense is to add absolute path representation to word embedding directly, and we also have tried this simple method. However, the overall conclusion of improvement diminishing still remains. Therefore, we believe that this phenomenon does not indicate that our proposed approach maybe be flawed. Instead, we think it is more likely due to these two kinds of paths' inherent properties.
>
> From Table3 presented in Section 5.2(Ablation Study), we conclude that after introducing full of the relative path encoding, the improvement from the absolute path encoding almost diminishes. We feel much sorry for that, because of lacking error bars for Table3, readers maybe deduce a misguided conclusion that a concrete conflict definitely exists between two kinds of paths. On the contrary, actually, the only thing we firmly conclude here is that these two kinds of path encoding methods are not orthogonal, which means overlap exists rather than conflict.
>
> For further explanation, we show the performance of TPTrans-alpha equipped with a pointer network (we repeat the experiment several times for confidence). Compared to TPTrans, TPTrans-alpha fully equips both the relative and absolute path encodings (Eq.7+Eq.10) (Ps: We are so sorry for only showing the results of Ruby and Js datasets due to tight time. And we will try our best to complete the results of the other two datasets getting enough time.), that is:
>
> |Model|Ruby.P|Ruby.R|Ruby.F1|JS.P|JS.R|JS.F1|
> |----|----|----|----|----|----|----|
> |TPTrans|32.40 $\pm$ 0.84|27.63 $\pm$ 0.78|29.82 $\pm$ 0.79|33.83 $\pm$ 0.55|28.37 $\pm$ 0.54|30.86 $\pm$ 0.53|
> |TPTrans-alpha|33.07 $\pm$ 0.86|28.34 $\pm$ 0.61|30.52 $\pm$ 0.53|33.68 $\pm$ 0.34|28.95 $\pm$ 0.54|31.14 $\pm$ 0.46|
>
>
> From this table, we conclude that although the improvement of TPTrans-alpha over TPTrans is not consistently significant, the so-called 'conflict is also not a certainty.
>
> We feel much sorry for any confusion. We will fix the misleading and even somewhat exaggerated explanation at L315-L319 and update the experiment results with error bars for Table3 in the later version.
>
>
>
>
> Q7: L320 "If two sub-tokens are too far away ... the relative and absolute path encoding play the same role ..." Can we find more evidence to justify or falsify this?
> Additionally, if there is any trimming of the paths (for example, due to input-length constraints), then absolute paths can keep information that would be otherwise discarded from (the longer) relative paths.
>
> A7: L320 says that if two source code tokens (sub-tokens for the case of the token split) are too far away from each other in the syntax tree, the relative path between them maybe span the root of a tree. For example, in Figure 1(b), please focus on those two tokens: the 'num' in 'def factorial(num):' and the 'result' in 'result=1'. Obviously, the relative path between these two tokens crosses the root of the syntax tree, which means that the relative path is equal to the concat of two tokens' absolute paths. Therefore, we think modelling relative or absolute paths can be considered equivalent in such a case.
>
> As you may concern, the relative path may be short (for example, one node exists in the path if two terminals have the same parent), but can also be long, and the max length is equal to twice the max absolute path length (similar to the case mentioned in the previous paragraph). To handle that, we set the max path length as 32, which covers almost all of the relative paths. Besides, for a few paths whose length exceeds 32, we adopt equal interval sampling to retain the information as much as possible. When the input data length exceeds the max, the usual approach in the NLP community is to truncate the last tokens. However, since the relative path represents the relationship between two tokens, the node at both the left and right sides of the path are equally important. For example, the leftmost and rightmost nodes in the relative path are the parents of these two terminals, respectively. Therefore, we adopt equal interval sampling instead of truncating last, and the pseudocode is:
>
>     'if len(path) > args.max_path_length:
>         idx = np.linspace(0, len(path) - 1, args.max_path_length, dtype=int).tolist()'.
>
> The sampling interval depends on the real length of the path. After that, we index nodes from the path and reassemble them.
>
> Finally, we want to emphasize again that the max path length 32 has already covered almost the relative paths, so the interval sampling can also be considered as the icing on the cake. So, let's answer your second question. The case you concern that relative path lost information due to trimming is almost non-existent.
>
> Q8: The efficacy of the proposed architecture is only evaluated over one training dataset and one task. The title says "for Code Representation" but the authors only experimented on code summarization.
>
> A8: Appreciate your criticism. As we all know, the method naming task is a popular and useful benchmark for code representation learning. Many works pick this task to show the ability of code representation learning[7][8]. In our paper, we experiment on four datasets across different languages to show our approaches' effectiveness. We're indeed happy to explore other tasks such as classification and retrieval, and we also have confidence in our proposed model.
>
>
> [1]Self-Attention with Relative Position Representations
>
> [2]A Survey on Knowledge Graphs: Representation, Acquisition and Applications
>
> [3]Translating Embeddings for Modeling Multi-relational Data
>
> [4]Rethinking the positional encoding in language pre-training
>
> [5]Attention is All Your Need
>
> [6]XLNet: Generalized Autoregressive Pretraining for Language Understanding
>
> [7]Language-Agnostic Representation Learning of Source Code from Structure and Context
>
> [8]Code2vec: Learning Distributed Representations of Code

---

### Review · Ethics_Reviewer_VZSi · 2021-07-22

**Recommendation:**

The two sections have some similarities but, as an outsider to the area, they just look like similar presentations of the same past related work. If so, I don't think it makes sense to ask the authors to change anything.

**Ethics Review:**

A reviewer suggested that the introductory material in Section 3 of this submission is possibly plagiarized from the introductory material in Section 3 of (https://arxiv.org/pdf/2103.11318.pdf).

---

### Review · Ethics_Reviewer_Uotp · 2021-08-10

**Recommendation:**

We recommend authors give proper reference to each and every equation/premise that is lifted from the existing papers.



**Ethics Review:**

The paper discusses the learning representation of source code using the transformer network and applies it for method name inference from the body of the source code.

First issue:
The section 3 (Approaches) first part indeed looks similar to the ICLR 2021 paper LANGUAGE-AGNOSTIC REPRESENTATION LEARNING
OF SOURCE CODE FROM STRUCTURE AND CONTEXT. Equation 1 in this paper has been lifted from the ICLR 21 paper without giving any reference.

Second issue:  Both papers have a very similar writing style and structure raising the question on anonymity.

---

> ### Author Response · Authors · 2021-08-19
> **Response**
>
> Thanks so much for pointing out our writing oversight. We re-examine Section 3 and indeed find that some sentences are quite similar to existing papers.  Actually, these parts are the presentation of the same past related works. We will carefully fix them in the later version. Thanks again for the kind reminder.

---

### Author Response · Authors · 2021-08-19
**One of the concerns about the difference between our model and the baselines**

Combining the two views about _Context_ and _Structure_ for source code has always been widely discussed in the research field of Code Intelligence. The _Context_ mainly refers to representing code as a sequence of text, and the _Structure_ refers to extracting knowledge from code snippets' syntax.

Unlike the Transformer-based baselines we picked, the Code2seq uses only pairwise path information in AST to represent code snippets. In other words, code2seq mainly focus on the code _Structure_ within the syntax tree and, as a result, lacks the modelling of _Context_. On the contrary, our proposed model and other Transformer-based baselines are designed based on Transformer, which means the _Context_ is naturally focused.

Essentially, the difference between our model and our baselines(Great, CodeTransformer) is encoding path vs encoding manually designed edges (Great) or distances/hops(CodeTransformer). The path's different node combinations contain plenty of structure information, which is overlooked by encoding distances. To verify the benefit of learning from paths than distance, we conduct ablation studies at L326-L340.

The XLNet is not designed for learning source code representation, and it is designed initially for huge-corpus natural language pretraining. The reason we list XLNet for comparison is that the Code Transformer is designed based on XLNet. Code Transformer additionally incorporates multiply structural relations within code syntax trees over XLNet. Although the XLnet is not specially designed for source code, its novel architecture of two-stream self-attention still shows powerful performance in code representation learning.

Please refer to the table below for clear comparisons.

|Model|Context|Structure|
|----|----|----|
|Code2Seq|No|Pairwise Path|
|Great|Yes|Manually Designed Edges|
|XLNet|Yes|No|
|CodeTransformer|Yes|Multiply Structural Distances|
|TPTrans|Yes|Pairwise Path|

---

### Author Response · Authors · 2021-08-21
**For the concern about the relationship between two kinds of paths**

1)Motivation:

we think it is meaningful to make an empirical study for the relationship between these two kinds of paths(called _relative_ path and _absolute_ path in our paper). Above all, lots of works have introduced either kind of path into many novel models[1-4]. However, to our best knowledge, none of them has assembled both paths simultaneously, and the relationship between these two kinds of paths is still unknown. On the other hand, as mentioned in our paper(L66-L70), both paths intuitively contain plenty of structural information of code syntax. For this reason, we think it is reasonable to focus on both of them instead of overlook either.

To further explain our empirical study's motivation, we also want to make an analogy from the research community of NLP. Technically, the ways we integrate path encodings into Transformer are mostly derived from the NLP field's _positional_ encoding methods. That is:

Relative path encoding vs. Relative positional encoding

Absolute path encoding vs. Absolute positional encoding

In the NLP community, the discussion about positional encoding has been going on for a long time, and tons of encoding method has been presented[5-7]. In contrast, as we mentioned above, the research of different paths is just emerging. From this point of view, we think our research is necessary and valuable for the Code Intelligence community.

2)Experiment Result and Conclusion

From Table3 presented in Section 5.2(Ablation Study), we conclude that after introducing full of the relative path encoding, the improvement from the absolute path encoding almost diminishes. It shows that these two kinds of paths are not orthogonal, which is still intuitive.

We feel much sorry for that, because of lacking error bars for Table3, readers maybe deduce a misguided conclusion that a concrete conflict definitely exists between two kinds of paths. Our somewhat misleading explanation at L319 further deepens it. On the contrary, actually, the only thing we firmly conclude here is that only feature overlap certainly exists rather than conflict.

For further explanation, we show the performance of TPTrans-alpha equipped with a pointer network (we repeat the experiment several times for confidence). Unlike TPTrans(Eq.6+Eq.7), TPTrans-alpha fully equips both the relative and absolute path encodings (Eq.7+Eq.10). The result is:

|Model|Ruby.P|Ruby.R|Ruby.F1|JS.P|JS.R|JS.F1|
|----|----|----|----|----|----|----|
|TPTrans|32.40 $\pm$ 0.84|27.63 $\pm$ 0.78|29.82 $\pm$ 0.79|33.83 $\pm$ 0.55|28.37 $\pm$ 0.54|30.86 $\pm$ 0.53|
|TPTrans-alpha|33.07 $\pm$ 0.86|28.34 $\pm$ 0.61|30.52 $\pm$ 0.53|33.68 $\pm$ 0.34|28.95 $\pm$ 0.54|31.14 $\pm$ 0.46|

|Model|Python.P|Python.R|Python.F1|Go.P|Go.R|Go.F1|
|----|----|----|----|----|----|----|
|TPTrans|38.76 $\pm$ 0.49|34.66 $\pm$ 0.35|36.59 $\pm$ 0.02|55.79 $\pm$ 1.3|51.05 $\pm$ 1.3|53.30 $\pm$ 0.83|
|TPTrans-alpha|38.39 $\pm$ 0.18|34.70 $\pm$ 0.26|36.45 $\pm$ 0.22|55.67 $\pm$ 0.92|51.31 $\pm$ 0.76|53.39 $\pm$ 0.13|

From this table, we conclude that the improvement of TPTrans-alpha over TPTrans is not consistently significant, and the so-called _conflict_ is also not a certainty. Actually, these results show that the feature overlap indeed exists. For these reasons, we finally select the TPTrans as our main proposed model, which is also consistent with the Principle of Occam's Razor.

We feel much sorry for any confusion. We will carefully fix them in the later version.


[1]Code2seq: Generating Sequences from Structured Representations of Code

[2]Code prediction by feeding trees to transformers

[3]Structural language models of code

[4]Code2vec: Learning Distributed Representations of Code

[5]Self-Attention with Relative Position Representations

[6]Rethinking the positional encoding in language pre-training

[7]Transformer-XL: Attentive Language Models Beyond a Fixed-Length Context

---

### Author Response · Authors · 2021-08-31
**Further experiments on Code Classification Task**

Since some reviewers concern that our proposed model is only evaluated on the code summarization task, we further experiment on the code classification task using the POJ-104 dataset[1], which contains 52000 C programs of 104 classes. We remove the Transformer decoder of our model and equip an attention-based weighted sum module to merge encoder features and then feed it into an MLP classifier.
Please note that:

1) most baselines of POJ classification use PycParser to parse code into AST while we still pick Tree-Sitter parser here;

2) unlike the standard experimental setup, many baselines omit each AST node's _Token_ field and only consider the _Type_ field to show the exclusive _structure_ learning power. For _approximate_ comparison, we mask all the variable names of CST, which means only  _Structure_ information is considered for our model. After that, only language Keywords are shown in the code sequence, and then the total vocab size reduces to 38. These results are present as POJ-Structure setting.

The baselines' accuracy:

|Setting|BiLSTM|Code2Vec|TBCNN|ASTNN|Treecaps|
|----|----|----|----|----|----|
|POJ|83.51|86.21|95.21|98.2|98.32|

|Setting|TBCNN|Treecaps|
|----|----|----|
|POJ-Structure|92.64|95.88|

And the performance of our models and ablation study:

|Model|POJ|POJ-Structure|
|----|----|----|
|TPTrans|94.94|93.03|
|w/o relative path in V|93.87|88.01|
|   w/o relative path in K,V|89.34|77.89|

First of all, we conclude that our model gets remarkable performance although not receives state of the art. The ablation study then shows that the path integrating method is still powerful on the classification task and improves much compared with the Vanilla Transformer (presented as 'w/o relative path in K,V').

Comparing the performance of our model with the baselines, we guess that the parser we use may not be the best choice for the code classification task. The authors of [2] also prove that different parsers can indeed result in much difference. So we think that the further research for different parsers would perhaps be valuable for the whole research community of code representation learning.

[1]Convolutional neural networks over tree structures for programming language processing

[2]TreeCaps: Tree-Based Capsule Networks for Source Code Processing

---

### Author Response · Authors · 2021-09-01
**The Further explanation about sequential information integrating**

In our paper, we says that _We show the feasibility of embedding sequential information as an inductive bias for the attention module of Transformers and are the first ones to do so to our best knowledge_ at L76. Here we explain further about this.

Above all, the _sequential_ information we say here means encoding a vector sequence, specifically referring to the path's node sequence in our paper. We firmly believe that integrating sequential information into Transformer as an inductive bias is not just trivial.

1)Firstly, directly integrating standard RNN into the Transformer's deep backbone most likely results in exploding or vanishing gradient.

2)Secondly, as we all know, the attention procedure of the Transformer can be seen as the similarity calculation for vectors. The _K_, _Q_ and _V_ of the Transformer's attention module all be normalized while the RNN's recursive process does not limit it, which may lead to the calculation instability for attention scores.

To overcome these issues, we carefully design the normalization method. Firstly, we replace the standard GRU cell as a layer-normalized GRU, leading to more stable parameters updates. Secondly, after getting the final state of a sequence, we feed it into a LayerNorm module to keep the same vector norm with _K_, _Q_ and _V_.


The result is:

|Model|Ruby.P|Ruby.R|Ruby.F1|JS.P|JS.R|JS.F1|
|----|----|----|----|----|----|----|
|TPTrans|32.40 $\pm$ 0.84|27.63 $\pm$ 0.78|29.82 $\pm$ 0.79|33.83 $\pm$ 0.55|28.37 $\pm$ 0.54|30.86 $\pm$ 0.53|
|  w/o Norm|26.91 $\pm$ 0.42|23.99 $\pm$ 0.31|25.37 $\pm$ 0.36|31.42 $\pm$ 0.16|26.34 $\pm$ 0.18|28.66 $\pm$ 0.10|


From this table, we conclude the normalization method is very effective. These results show that it is harder for the composite model to fully extract _sequential_ information without norm control.

So we say that we prove the feasibility of embedding sequential information as an inductive bias for the attention module of Transformers. And we think that our approach also contributes to the design of _neural network architectures_, not only to the community of Code Intelligence.

We have mentioned this design in L254, although many details still have not been presented in our paper. But we are also glad to add the details in the later version if reviewers suggest it.

---

### Decision · Program_Chairs · 2021-09-28

**Decision:**

Accept (Poster)

**Comment:**

The reviewers agreed that even though the general aspects of using recurrent units for embedding paths, and using relative and absolute paths for embedding source code are not completely novel, the paper presents a nice novel combinations of these ideas. There were many clarity issues in the writing, but the response helped clarify many of them. One of the major concerns that came up was that evaluation did not clearly show that the proposed architecture was consistently better, and the evaluation was only performed on a single task of code summarization. The new results for code classification task are interesting, but it is not clear if TPTrans performs better than stronger baselines. Finally, there were also some concerns regarding better understanding of the benefits and limitations of the approach, and how incorporating paths might improve (or worsen) the performance of the model.

**Consistency Experiment:**

NeurIPS has a long history of experimentation. In 2014, NeurIPS ran an experiment in which 10% of submissions were reviewed by two independent committees to quantify the randomness in the review process. This year, we repeated a variant of this experiment to see how the quality of the review process has changed over time.  This paper was part of the experiment and was therefore assigned to two committees (consisting of reviewers, an Area Chair, and a Senior Area Chair) that reached independent decisions.  If both committees made the same recommendation, this recommendation was followed. If a single committee recommended acceptance, the paper was accepted (with the exception of a few cases in which the other committee identified what we considered a fatal flaw, e.g., an error in a key result).

This copy’s committee reached the following decision: **Reject**

The other committee assigned to the paper recommended **Accept (Poster)**.  You can find the other set of reviews, along with any follow up discussion with the authors here:
https://openreview.net/forum?id=70Q_NeHImB3